# Spatiotemporal identification of druggable binding sites using deep learning

Igor Kozlovskii [1] & Petr Popov[1 ✉]

Identification of novel protein binding sites expands druggable genome and opens new opportunities for drug discovery. Generally, presence or absence of a binding site depends on the three-dimensional conformation of a protein, making binding site identification resemble the object detection problem in computer vision. Here we introduce a computational approach for the large-scale detection of protein binding sites, that considers protein conformations as 3D-images, binding sites as objects on these images to detect, and conformational ensembles of proteins as 3D-videos to analyze. BiteNet is suitable for spatiotemporal detection of hard-to-spot allosteric binding sites, as we showed for conformation-specific binding site of the epidermal growth factor receptor, oligomer-specific binding site of the ion channel, and binding site in G protein-coupled receptor. BiteNet outperforms state-of-the-art methods both in terms of accuracy and speed, taking about 1.5 minutes to analyze 1000 conformations of a protein with ~2000 atoms.

[1] iMolecule, Center for Computational and Data-Intensive Science and Engineering, Skolkovo Institute of Science and Technology, Bolshoy Boulevard 30, bld. 1, Moscow 121205, Russia. ✉email: p.popov@skoltech.ru

Proteins serve biological functionality of a cell via local intermolecular interactions that take place in spatial regions, called binding sites. Binding sites are one of the key elements in drug discovery, being hot spots in the pharmacological targets, where the designed drug-like molecule should bind. Identification of novel binding sites expands druggable genome and opens new strategies for therapy and drug discovery[1]. Typically drug-like molecules target either orthosteric binding site, where protein interacts with endogenous molecules, or topologically distinct allosteric binding sites[2]. The latter is of a special interest, because allosteric binding sites exhibit higher degree of sequence diversity between protein subtypes, thus, allowing to design more selective ligands, in contrast to the orthosteric ligands[3–5].

Proteins are flexible molecules, that adopt various conformations during their life cycle; and a binding site is a dynamic property of a protein mediated by its conformational changes[6,7]. Single protein structure represents only a minor part of the entire conformational space, hence, binding sites might be easy to overlook from the experimentally determined three-dimensional protein structures[8,9]. Moreover, many proteins perform their function assembling to oligomeric structure and can form binding sites by means of oligomer's subunits[10,11].

Experimental identification of binding sites, such as fragment screening and site-directed tethering[12,13], using antibodies[14], small molecule microarrays[15], hydrogen-deuterium exchange[16], or site-directed mutagenesis[17] are resource-consuming and may result in negative outcome. On the other hand, computational methods allow to perform large scale binding site identification, investigate protein flexibility via molecular dynamics simulation, and probe to fit chemical compounds using virtual ligand or fragment-based screening. The classical approaches typically employ empirical scoring functions based on the structural information about known binding sites, or use this information as features for the machine learning algorithms[18–28]. The success rate of these approaches critically depends on the designed features, and may result in false positive predictions, that is identification of undruggable regions[29]. Most recently, deep learning approaches, that do not require hand-crafted feature engineering, demonstrated feasibility to predict protein binding sites[30]. In spite of present progress, large-scale binding site detection remains to be a challenge, let alone that there is still a big room for improvement in terms of the method's accuracy[28].

In this study, we present rapid and accurate deep learning approach, dubbed BiteNet (**Bi**nding **s**ite neural **Net**work), suitable for the large-scale and spatiotemporal identification of protein binding sites. Inspired by the computer vision problems, such as object detection in images and videos, we consider protein conformations as the 3D images, binding sites as the objects on these images to detect, and conformational ensembles, that is a set of protein conformations, as the 3D videos to analyze. We showed that BiteNet is capable to solve challenging binding site detection problems by applying it to three-dimensional structures of pharmacological targets, including ATP-gated cation channel, epidermal growth factor receptor, and G protein-coupled receptor. Particularly, BiteNet correctly identified oligomer-specific allosteric binding site formed by the subunits of the trimeric P2X3 receptor complex; and conformation-specific allosteric binding site of the epidermal growth factor receptor kinase domain. BiteNet can be used for spatiotemporal investigation of novel binding sites, as we showed by the example of molecular dynamics simulation trajectory for the adenosine A2A receptor. BiteNet outperforms the state-of-the-art methods both in terms of accuracy and speed as demonstrated on several benchmarks. It takes approximately 0.1 seconds to analyze single conformation and 1.5 minutes for BiteNet to analyze molecular dynamics

trajectory with 1000 frames for protein with ~2000 atoms, making it suitable for large-scale spatiotemporal analysis of protein structures.

## Results

**BiteNet architecture.** To develop BiteNet we trained 3D convolutional neural network using manually curated protein structures from the Protein Data Bank as the training set (see "Methods" section). Figure 1 presents the BiteNet workflow. Similarly to 2D images, that have two dimensions (width and height) and three channels for each pixel (red, green, and blue), we represent proteins as 3D images with three dimensions (width, height, and length) and 11 channels for each voxel, where channels correspond to the atomic densities of a certain type (see "Methods" section) (Fig. 1a). As neural networks typically take fixed size tensors for the input, we used voxel grid of $64 \times 64 \times 64$ voxels and voxels of $1 \text{ Å} \times 1 \text{ Å} \times 1 \text{ Å}$ size. If protein exceeds 64 Å in any of the dimensions, we used several voxel grids to represent it (Fig. 1b). The obtained voxel grids are processed with the 3D convolutional neural network (Fig. 1c) to output $8 \times 8 \times 8 \times 4$ tensor, where the first three dimensions correspond to the cell coordinates relatively to the voxel grid (region of $8 \times 8 \times 8$ voxels), and the four scalars of the last dimension correspond to the probability score of the binding site being in the cell and its Cartesian coordinates. This is followed by the processing of the obtained tensors to output the most relevant predictions of the binding sites (Fig. 1d). Thus, the input to the BiteNet is the spatial structure of a protein and the output is the centers of the predicted binding sites along with the probability scores. Finally, BiteNet identifies the amino acid residues of a binding site within 6 Å neighborhood with respect to the predicted center. Additionally, when applied to the conformational ensemble of a protein, the obtained predictions and identified amino acid residues are grouped using clustering algorithms (see "Methods" section).

**Spatiotemporal prediction of binding sites in pharmacological targets.** To demonstrate applicability of BiteNet we considered challenging binding site detection problems comprising three pharmacological targets: the P2X3 receptor of the ATP-gated cation channel family, the epidermal growth factor receptor of the kinase family, and the adenosine A2A receptor of the G-protein coupled receptor family.

*ATP-gated cation channel.* The ATP-gated cation channel, formed by the P2X3 receptor, mediates various physiological processes and represents pharmacological target for hypertension, inflammation, pain perception, and others[31]. The channel consists of three identical monomers traversing the membrane, and the orthosteric ATP-binding site comprises amino acid residues of two monomers (see Fig. 2c)[32]. Drug design targeting the orthosteric binding site is difficult due to highly polarized ATP-specific interface, on the other hand, allosteric ligands targeting protein–protein interactions form promising avenue for drug discovery[11]. Recently allosteric binding site formed by two monomers of a channel was discovered for the P2X3 and P2X7 receptors[11,33]. We applied BiteNet to the ATP-bound and (AF-219)-bound structures of the trimer complex formed by the P2X3 monomers (PDB IDs: 5SVK, 5YVE), as well as to the single monomer structures. BiteNet correctly identified the orthosteric binding site in the ATP-bound structure and the allosteric binding site in the (AF-219)-bound structure of the trimer, and not in the monomer structures (see Fig. 2). Interestingly, BiteNet also predicted center for the ATP-binding site located on the opposite end of the ATP molecule with lower

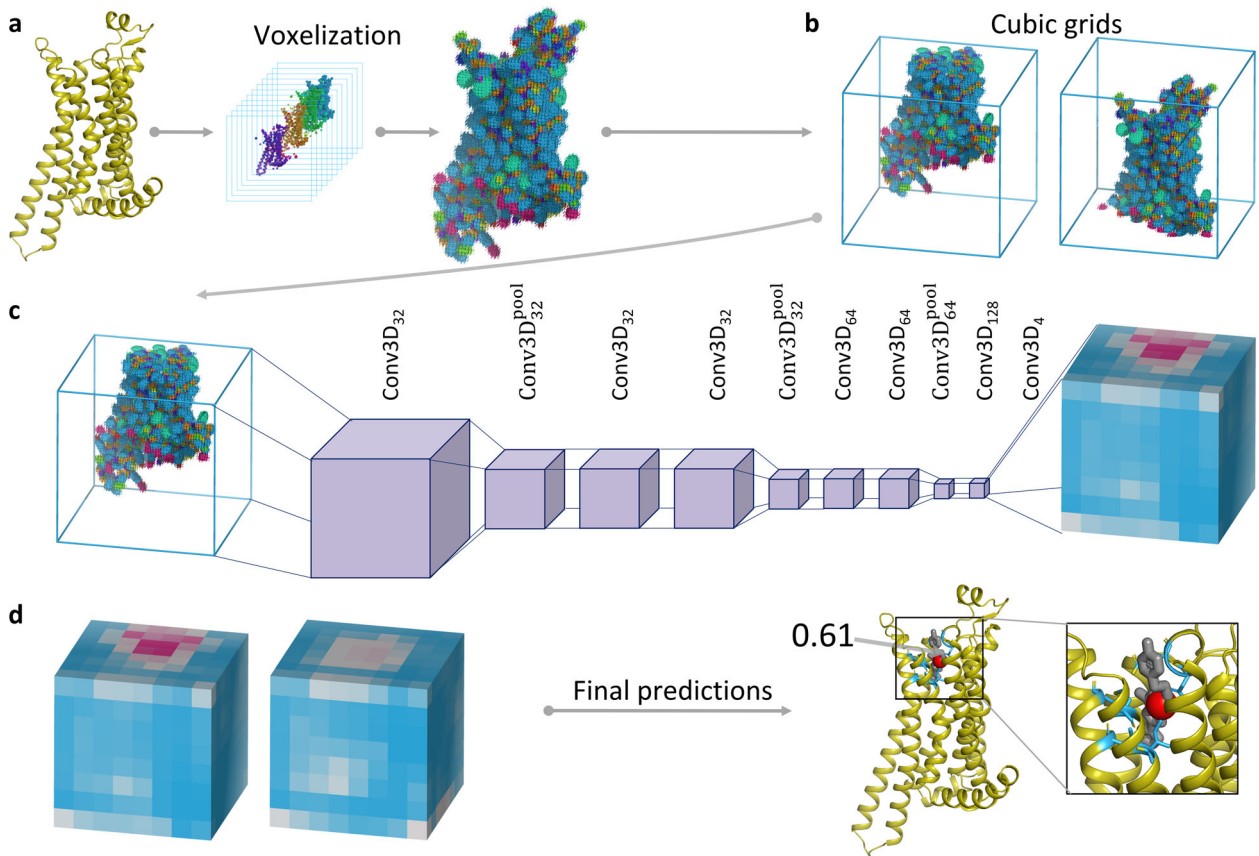

**Fig. 1 Schematic representation of the BiteNet workflow. a** The input three-dimensional structure of a protein is represented with voxel grid, where channels correspond to the atomic densities. **b** The voxel grid is split into fixed-size cubic grids to be fed a neural network. **c** Each cubic grid is processed with the 3D convolutional neural network to predict binding sites in fixed-size cells. Cells in cubic grids are colored according to the probability score confidence, from blue to red. **d** Predictions obtained for each cubic grid are then processed to output center of the binding site (red sphere), its probability score, and amino acid residues within 6 Å of the predicted center (blue sticks). Co-crystallized ligand is shown with gray sticks.

probability score (see Supplementary Fig. 1). To ensure, that this is not an artifact of the rotational variance of the model, we generated 50 replicas by rotating the monomer about ten axes by $\pi/3$, $2\pi/3$, $\pi$, $4\pi/3$, and $5\pi/3$ angles and averaged the obtained predictions. As one can see from Fig. 2e, f although the absolute values of the probability scores vary with respect to the monomers, in all the cases BiteNet correctly identifies the allosteric binding site for the trimer complex and not for the monomer. Note, that ATP is endogenous agonist, while AF-219 is antagonist for the P2X trimer. The agonist-bound and the antagonist-bound conformations are different, particularly, in the regions of the orthosteric and allosteric binding sites (Fig. 2c, d). Therefore, BiteNet is sensitive to the conformational changes, as it does not predict the ATP-binding site in the (AF-219)-bound structure and vice versa. Interestingly, despite absence of binding site in the monomer structure, BiteNet predicted different binding sites with relatively high score in the monomer structures. Closer look into available three-dimensional structures of the P2X3 receptors revealed cation ions (Mg, Na, Ca), and ethylene glycol molecules corresponding to these predictions (PDB IDs: 5YVE, 5SVS, 5SVT, 5SVJ, 5SVR, 5SVQ, 5SVP, 5SVM, 5SVL, 6AH4, and 6AH5). We would like to emphasize that the training set does not contain structures similar to the P2X3 receptor. Indeed, the maximal sequence identity is 0.32 for human heparanase (PDB ID: 5L9Z) and the maximal structure similarity is 0.6 for tyrosine carboxypeptidase (PDB ID: 6J4P). Thus, this case demonstrates

predictive power of BiteNet, rather than detection of memorized binding sites.

*Epidermal growth factor receptor (EGFR).* EGFR is a transmembrane protein from the tyrosine kinase family. Over-expression of EGFR is associated with various types of tumors. Although there are EGFR inhibitors targeting the orthosteric binding site of the kinase domain, proteins found in cancer cells often have amino acid substitutions making it insensitive to such inhibitors. There are also mutant-selective irreversible inhibitors that covalently bind to the Cys797 amino acid residue, however, some mutant type receptors possess different amino acid residue at 797 position as well[34]. Recently, three-dimensional structure of L858R/T790M EGFR kinase domain variant bound to the mutant-selective allosteric inhibitor EAI001 was discovered (PDB ID: 5D41)[35]. It was shown, that EAI001 binds to only one monomer, leading to incomplete inhibition, but decreasing cell autophosphorylation. Accordingly, the three-dimensional structure is asymmetric dimer with one monomer bound to both orthosteric and allosteric ligands (the ATP-analog adenylyl-imidodiphosphate (AMP-PNP) and EAI001, respectively), while the other monomer bound to AMP-PNP only. BiteNet successfully identified both orthosteric and allosteric binding sites in one monomer (chain A) and only former in the other monomer (chain B). In contrast to the P2X3 case study, the training set does contain two EGFR kinase domain structures (PDB IDs: 5UG9, 5GNK) as well as three other proteins with high sequence identity and

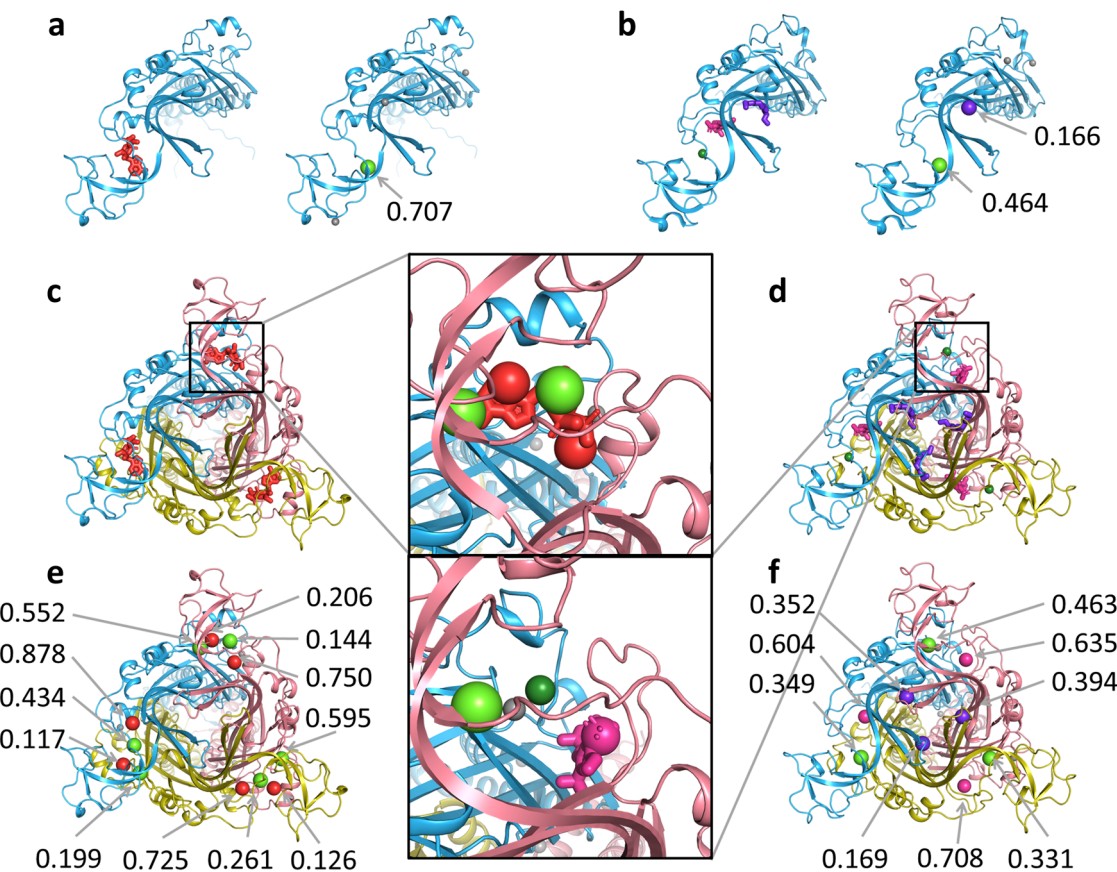

**Fig. 2 BiteNet predictions for the monomer and oligomer structure of the P2X3 receptor. a** Monomer structure with the orthosteric ligand and cation ion alongside the BiteNet prediction for this structure. **b** Monomer structure with the allosteric ligand, cation ion and ethylene glycole alongside the BiteNet prediction for this structure. **c, d** Agonist-bound and antagonist-bound structures of the P2X3 trimer, respectively. **e, f** BiteNet predictions for the agonist-bound and antagonist-bound structures of the P2X3 trimer, respectively. Orthosteric and allosteric ligands are shown with red and magenta sticks, respectively. cation ions are shown as dark green spheres and ethylene glycol molecules are shown with violet sticks. BiteNet predictions for these molecules are shown as spheres with the corresponding color.

structure similarity: DDX25 RNA helicase (PDB ID: 2RB4), kinase domain of human HER2 (PDB ID: 3PP0) and HER3 pseudokinase domain (PDB ID: 4OTW) with sequence similarity of 0.722, 0.762, and 0.596 and structure similarity 0.833, 0.876, and 0.944, respectively. Nonetheless, all these structures have ligands bound to the binding sites corresponding to the EGFR orthosteric binding site, but not to the allosteric binding site. Therefore, this example shows the predictive power of BiteNet to detect conformation-specific binding sites.

Although this and previous examples clearly demonstrate BiteNet's capability to detect binding sites in *holo* conformations, on practice, such conformations can be unknown, especially, when one wants to discover novel binding sites. To evaluate BiteNet's ability to detect binding sites starting from the unbound conformation, we emulated unbound-to-bound conformational transition as it follows. First, we modeled missing residues in chain B and placed EAI001, as it is observed in chain A. Then, we prepared molecular dynamics system containing chain B, AMP-PNP and EAI001, embedded into the water box with ions using the CHARMM-GUI web server[36]. Next, we run full atom energy minimization of the prepared system until convergence using Gromacs[37], resulting in minimization trajectory consisting of ~900 conformations. Finally, we removed ligands, ions, and water and applied BiteNet to each frame of the minimization trajectory along with its 50 replicas. Figure 3c shows, that the probability score for the allosteric binding site steadily increases, while the energy of the system is decreasing and the root mean square deviation (RMSD)

with respect to the allosteric binding site in the starting (unbound) conformation is increasing. Supplementary Movie 1 (Fig. 4a) demonstrates BiteNet predictions along with the minimization trajectory. Note, that the probability score for the orthosteric binding site remains high during the minimization. Also note, that we used 4Å for the non-max suppression distance threshold in order to avoid merging of the predictions for orthosteric and allosteric binding sites during post-processing stage of BiteNet. Therefore, BiteNet can be applied for the large-scale spatiotemporal trajectories in order to detect protein conformations that possess binding sites unseen in the original structure.

*G protein-coupled receptor (GPCRs).* GPCRs mediate numerous physiological processes in the body, making them important targets for modern drug discovery. Most of FDA-approved drugs bind to orthosteric binding sites of GPCRs. However, such drugs may be nonselective with respect to the highly homologous receptor subtypes. In such cases, there is need in drug design targeting allosteric binding sites, that are less conserved than orthosteric one[38]. Three-dimensional structures of GPCRs reveal allosteric binding sites spanning extracellular, transmembrane, and intracellular regions; identification of novel allosteric sites in GPCRs can provide alternative options for drug discovery[39]. To demonstrate the use of BiteNet in spatiotemporal identification of GPCR binding sites we analyzed molecular dynamics trajectories of the human adenosine A2A receptor (A2A) retrieved from the GPCRmd repository[40].

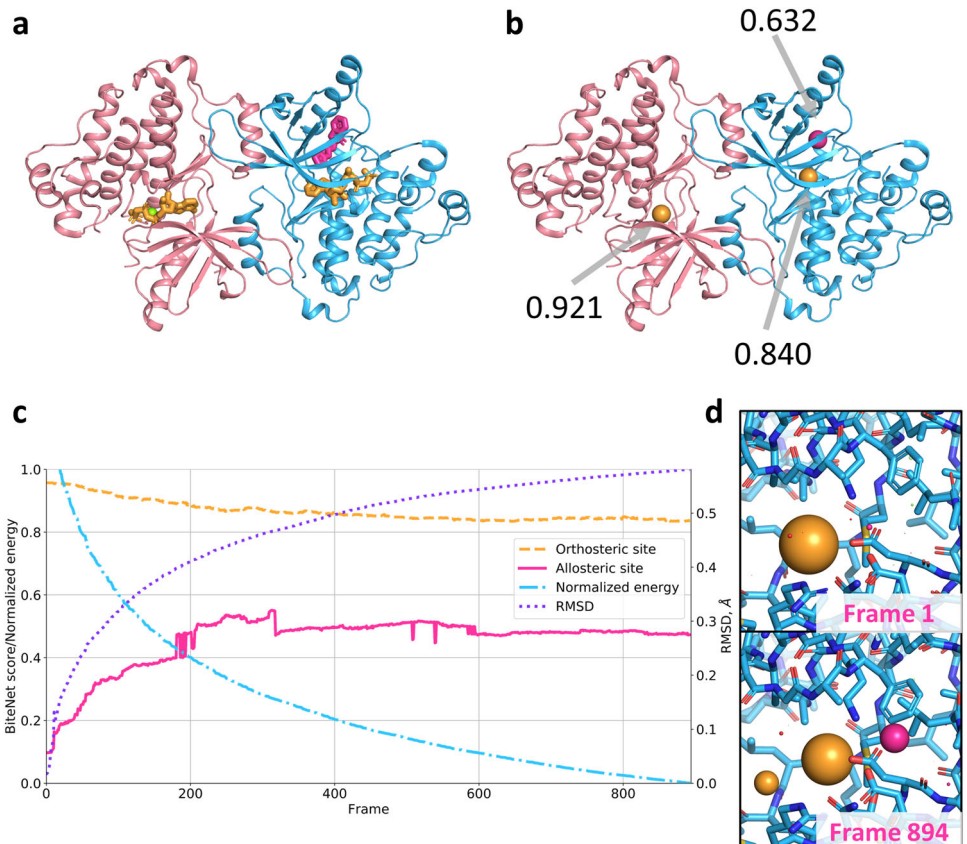

**Fig. 3 BiteNet predictions for the energy minimization trajectory of the assymetric dimer structure of the EGFR kinase domain. a** Assymetric dimer structure of the EGFR kinase domain. Orthosteric and allosteric ligands are shown with yellow and magenta sticks, respectively, Mg ion is shown as green sphere. **b** BiteNet predictions for the assymetric dimer, the predicted centers for the ligands are shown as spheres with the corresponding color. **c** BiteNet predictions obtained for the energy minimization trajectory. The normalized energy is shown with blue dash-dotted line, the RMSD with respect to the unbound conformation of the alloteric binding site is shown with violet dotted line, BiteNet probability score for the orthosteric and allosteric binding sites are shown with dashed orange and magenta solid lines, respectively. The normalized energy of 1 and 0 corresponds to $-7.76969 \times 10^{5}$ kJ/mol and $-8.80655 \times 10^{5}$ kJ/mol, respectively. **d** The starting and the final conformations of the minimization trajectory along with BiteNet predictions.

Namely, we considered trajectories of A2A embedded into the POPC lipid bilayer surrounded by water, sodium and chloride ion molecules starting from the active-like conformation (PDB ID: 5G53) in complex with agonist NECA and with no ligand (GPCRmd IDs: 48:10498 and 47:10488, respectively). In total each simulation lasted for 500 ns with the time step of 4.0 fs and interval between frames of 2.0 ns, resulting in 2500 conformations of A2A. We consequently applied BiteNet for each frame of the trajectory. As expected, in both simulation trajectories we observed a cluster of predictions corresponding to the canonical orthosteric binding site in GPCRs. The cluster is more dense and with higher averaged score in the ligand-bound simulation trajectory, which could be explained by lower flexibility of the protein due to the protein–ligand interactions. Surprisingly, in both simulation trajectories we also observed cluster of predictions in the neighborhood of the end of TM1, TM7 and helix 8 starting from ~300 ns in the ligand-free simulation and from ~150 to ~200 ns and from ~320 to 370 ns in the ligand-bound simulation. Closer look to the conformations with the highest probability scores corresponding to this cluster revealed lipid tail buried to the cavity formed by hydrophobic amino acid residues. It is important to note, that although GPCRs are tightly surrounded by lipids, BiteNet did not produced predictions all over the region exposed to a membrane, as it was explicitly trained on druggable binding sites. To investigate if the lipid tail binds to the cavity, for each frame $f$ we calculated its mobility in terms of RMSD between the conformation of the lipid tail in this frame and the conformation of the lipid tail averaged over [$f - 100, f + 100$] frames. As one can see from Fig. 5c, d, the calculated RMSD is lower for the frames with high probability scores corresponding to the predicted binding site. Supplementary Movies 2 and 3 (Fig. 4b, c) demonstrates BiteNet predictions and binding of the POPC molecule during these simulations. To the best of our knowledge there is no available structures for any GPCR with ligand bound to this region. When applied BiteNet to molecular dynamics trajectories obtained for other receptors from GPCRmd, we also observed similar cluster in the muscarinic M2 receptor, again, starting from active-like conformation. Thus, the predicted region may be worth paying attention to, as it may correspond to the novel allosteric binding site in GPCRs.

To summarize, we showed applicability of BiteNet for binding site detection for three different pharmacological targets and challenging binding sites observed in soluble as well as in transmembrane protein domains. BiteNet was capable to detect conformation-specific and oligomer-specific allosteric binding sites and can be applied for large-scale spatiotemporal analysis of protein structures. Using the example of A2A we demonstrated how BiteNet can be used on practice to investigate novel binding sites. We also would like to note, that used three-dimensional structures were not exposed to BiteNet during the training process. In the next section, we demonstrate computational efficiency of BiteNet in terms of accuracy and speed by

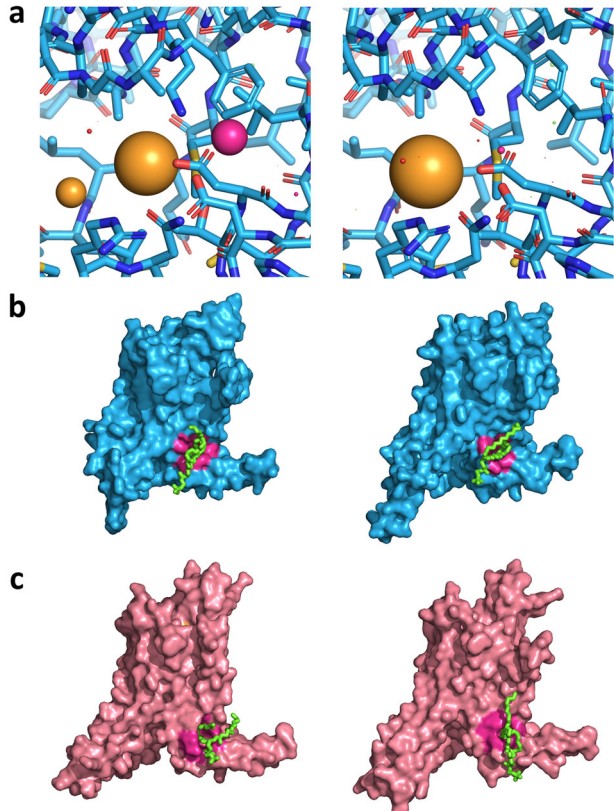

**Fig. 4 Video frames of energy minimization and molecular dynamic trajectories analyzed with BiteNet. a** BiteNet applied to the minimization trajectory of the EGFR kinase domain starting from the unbound state (Supplementary Movie 1). Predictions corresponding to the orthosteric and allosteric sites are shown as yellow and magenta spheres, respectively. Frames 1 and 894 are shown. **b, c** BiteNet applied to the ligand-free (**b**) and ligand-bound (**c**) A2A molecular dynamics trajectory (Supplementary Movies 2 and 3, respectively). BiteNet predictions for the orthosteric and hypothethical binding sites are colored with yellow and magenta, respectively. Lipid molecule, that occupies the identified binding site, is shown with green sticks. Frames 1489 and 2055 are shown for the ligand-free simulation, and frames 835 and 1806 are shown for the ligand-bound simulation.

comparing it against the existing computational methods on binding site prediction benchmarks.

**Computational efficiency of BiteNet**. To compare BiteNet with the other approaches we evaluated its performance on the HOLO4K and COACH420 benchmarks (see "Methods" section). As the performance metric we used the average precision (AP), as we consider this metric the most suitable for the binding site prediction problem (see "Discussion" section). We calculated AP for *All* and *TopN* predictions, where $N$ is the number of the true binding sites present in a protein structure. As one can see from Fig. 6a BiteNet significantly outperforms ($p$-value $\leq 1.2e^{-6}$) classical binding site prediction methods, such as fpocket[23], SiteHound[21], MetaPocket[24], as well as the state-of-the-art machine learning methods, such as DeepSite[30] and P2Rank[28] (Supplementary Tables 3–12 lists more detailed comparison including the performance on the entire benchmarks, as well as the precision, recall, true positive, false positive, and false negative metrics).

BiteNet is also computationally efficient, Fig. 6b shows elapsed time spent by BiteNet along with fpocket and P2Rank, which are

one of the fastest methods, with respect to the number of the processed protein conformations. BiteNet, that runs on a single GPU (GeForce GTX 1080 Ti), outperforms P2Rank that runs on several CPUs (Intel(R) Core(TM) i7-8700K CPU @ 3.70 GHz). On average, BiteNet takes approximately 0.1 seconds to process single protein conformation. Further optimization of CPU–GPU interconnection and multiple GPUs implementation of BiteNet will result in even faster performance.

We observed that BiteNet's performance is 5% higher, when the true positive prediction of a binding site is defined as in the training, as compared to the P2Rank's criterion. The main reason for this is more strict ligand filtering implemented in the training aiming to discard not relevant small molecules. For example, in the HOLO4K benchmark there are 29 structures corresponding to the Aspartic peptidase A1 protein family, that have the only active site with a peptide-like molecule bound to it. However, there are also small sugars (mannoses or arabinoses) that surround protein structures yielding additional binding sites according to the P2Rank's criterion (see Supplementary Fig. 2). Therefore the total number of binding sites is 29 for the BiteNet's criterion and 38 for the P2Rank's criterion. As a result, BiteNet yields zero false negative predictions in the former case, and nine in the latter case, hence, the drop in the AP metric from 0.99 to 0.75.

To investigate BiteNet's predictive power in more detail we considered its performance on the most represented protein families comprising the HOLO4K benchmark retrieved from the InterPro database[41]. More precisely, we assigned the InterPro family identifier to each protein in the HOLO4K benchmark and considered protein families counting at least 20 protein structures and containing at least one relevant binding site according to the BiteNet's criterion. Figure 7 shows the AP metric calculated for each protein family for BiteNet, as well as the ratio of structures from this family presented in the training set. BiteNet outperforms the other methods on 17 out of 27 protein families, for two protein families BiteNet is on par with P2Rank showing the perfect performance, and for the rest eight protein families there is a method with better performance than BiteNet (see Supplementary Fig. 3). Note also that none of the protein families are over-represented in the training set (the median ratio is 0.15%). Figure 8 demonstrates common types of false positive and false negative predictions on example of Glycosyl transferase protein family (IPR000811). The most common false positive predictions correspond to the ligand-free region with low probability score ($\leq 0.15$) (see Fig. 8a). Interestingly, another type of false positive predictions correspond to the region with absent ligand in one structures, but present in the others (see Fig. 8c). Given higher probability scores ($\geq 0.20$) and capability to bind a ligand to the predicted binding site in some protein structures, it is not clear whether these predictions should be considered as false positive. At the same time, there are structures with the bound ligand, but with no binding site predictions, corresponding to the most common type of the false negative predictions (see Fig. 8b). Finally, we observed that some false negative predictions correspond to ligands in the proximity of the catalytic binding site predicted with high probability score ($\geq 0.75$) for the PLP molecule (pyridoxal-5′-phosphate) (see Fig. 8d). Thus, such false negative predictions might be an artifact of the nonmax-suppression procedure, when only single prediction with the highest probability score is kept within the 8 Å.

## Discussion

In this study we introduced BiteNet, a deep learning approach for spatiotemporal identification of binding sites. BiteNet takes advantages of the computer vision methods for object detection, by representing three-dimensional structure of a protein as a 3D

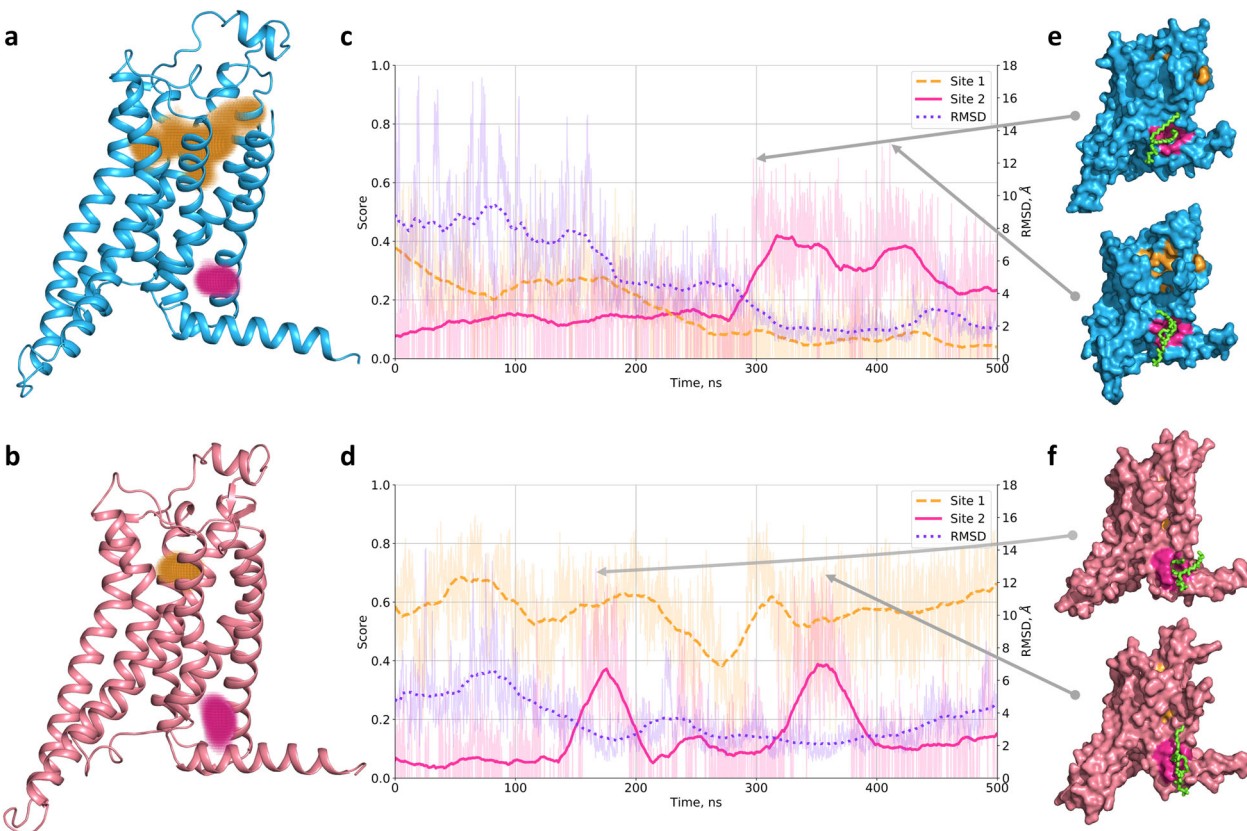

**Fig. 5 BiteNet predictions for molecular dynamics trajectories of the adenosine A2A receptor. a, b** Starting ligand-free and agonist-bound conformations of A2A, respectively. Orange point clouds corresponds to the BiteNet predictions of the canonical orthosteric binding site in A2A, while magenta point cloud corresponds to the BiteNet predictions of the hypothetical binding site, observed during the simulation. **c, d** BiteNet probability scores for the orthosteric binding site (dashed orange line), allosteric binding site (magenta solid line), and RMSD with respect to the window-based mean lipid tail conformation (dotted violet line), computed for the molecular dynamics trajectories. **e, f** A2A conformations corresponding to the highest BiteNet probability scores for the hypothetical binding site. Lipid molecule, that occupies the hypothetical binding site, is shown with green sticks.

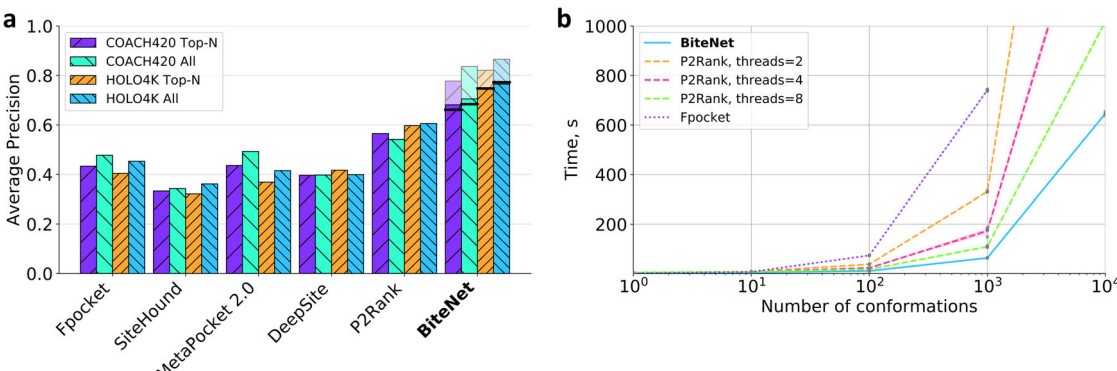

**Fig. 6 Predictive power and computational efficiency of BiteNet. a** Performance of the binding site prediction methods on the COACH420 and HOLO4K benchmarks. Violet and orange bars with diagonal hatching correspond to the average precision calculated for top *N* predictions for the COACH420 and HOLO4K benchmarks, respectively, where *N* is the number of true binding sites in a protein. Similarly, cyan and blue back hatched bars correspond to the average precision calculated for all predictions for the COACH420 and HOLO4K benchmarks, respectively. Pale bars correspond to the BiteNet performance, when the true positive binding site is defined as in the training. Black lines correspond to the BiteNet performance on the whole benchmarks. **b** Elapsed time for fpocket (dotted violet line), P2Rank (dashed orange, magenta, and green lines), and BiteNet (solid blue line) to analyze 1, 10, 1000, and 10,000 conformations of a protein with ~2000 atoms. The computed elapsed time is the average of ten independent runs, individual data points are shown with gray circles.

image with channels corresponding to the atomic densities. BiteNet goes beyond classical problem of binding site prediction in *holo* protein structures, exploring protein dynamics and flexibility by means of large-scale analysis of conformational

ensembles. The detected conformations with observed binding site of interest, then can be used for structure-based drug design approaches, such as molecular docking and virtual ligand screening, as well as structure-based de novo drug design.

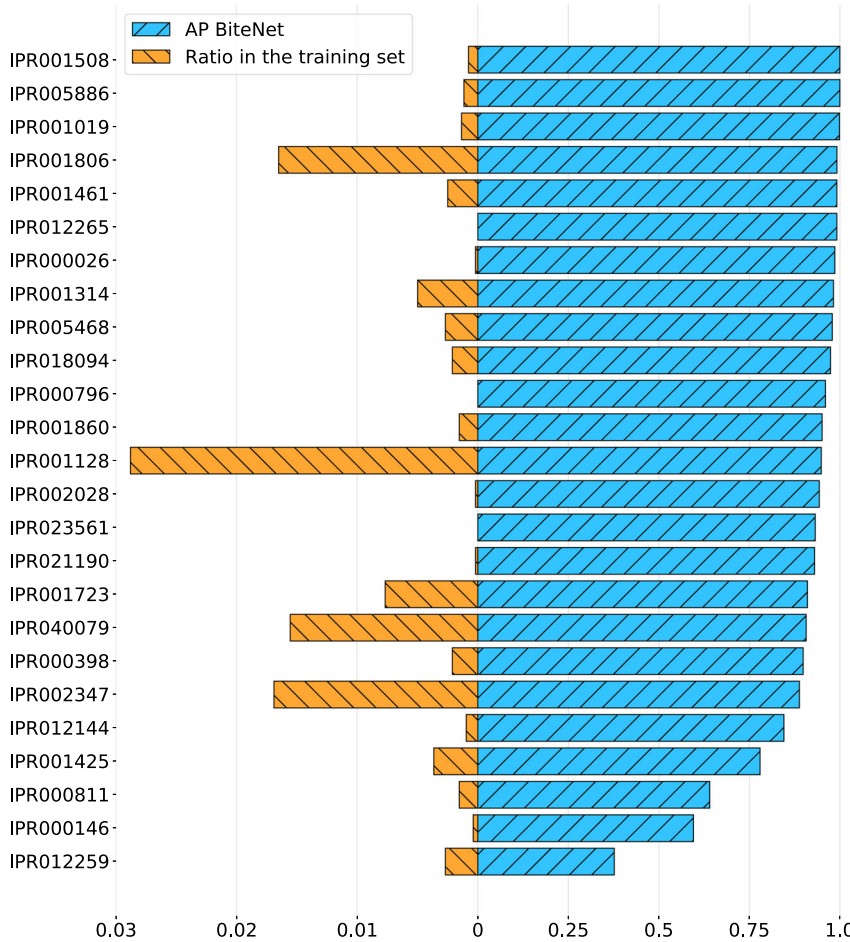

**Fig. 7 BiteNet performance on the most representative protein families in the HOLO4K benchmark.** Average precision calculated for protein families with at least 20 protein structures in the HOLO4K benchmark is shown with diagonal hatched blue bars. Ratio of structures from each protein family presented in the training set is shown with back diagonal hatched orange bars.

We believe superior performance of BiteNet with respect to the other machine learning methods for binding site prediction was achieved due to careful preparation of the training set and training process; below we address several important issues related to these procedures.

Curated and well-balanced training set is of crucial importance for derivation of machine learning models and its applicability domain. Experimentally determined protein structures often contain detergent and buffer molecules, that reveal electron density. This should be considered carefully and not mixed up with the true binding sites. To avoid potential bias related to this problem we filtered out typical detergent and buffer molecules (see Supplementary Table 1). Note, however, this procedure likely resulted in removing both false and true positives binding sites. For example, we discarded lipid molecules surrounding membrane proteins, including functional lipid molecules, such as cholesterol. Additionally, training set inevitably contains false negative binding sites, because protein structures may also contain empty binding sites. Another source for false positive binding sites come from symmetrical oligomer structures, as for example the P2X3 trimer. Indeed, the asymmetric unit does contain the ligand, however the binding site is formed not only by the asymmetric unit, but also by symmetry mates, which are usually omitted in the analysis. We also observed structures with missing atoms and residues in the binding sites; we believe such structures should be either properly refined or discarded from the training set. In addition, the definition of the true positive prediction and

binding site itself may vary. Binding site is typically defined with respect to the cutoff distance between the protein and ligand atoms (4.0 Å in this study), center of the binding site can be defined as the center of mass of the ligand or the binding site residues (in this study), and the true positive prediction can be defined with respect to the cutoff distance between the ligand or center of the binding site (4.0 Å in this study). We choose the latter definition of the true positive prediction because it is invariant with respect to the type of the ligand and its binding pose.

Training-validation split is another important issue, that affects performance of the derived model. First of all structural similarity should be taken into account, as it is known that proteins with low sequence similarity may still share highly similar protein fold. We observed that the largest cluster contains 4044 protein chains of similar structures. Splitting this cluster into the train and validation sets would likely result in the bias and overfit with respect to the corresponding protein fold. To circumvent this issue we carefully distributed protein structures, such that there is no highly similar structures in the training and validation sets in terms of the TM-score structural similarity[42].

Data augmentation techniques can be also helpful to derive more robust predictive models. For protein binding site prediction problem, computational methods to generate conformational ensembles can be used in order to represent binding site with multiple orientations or even small perturbations. In this study, due to computational limitations, we used implicit data

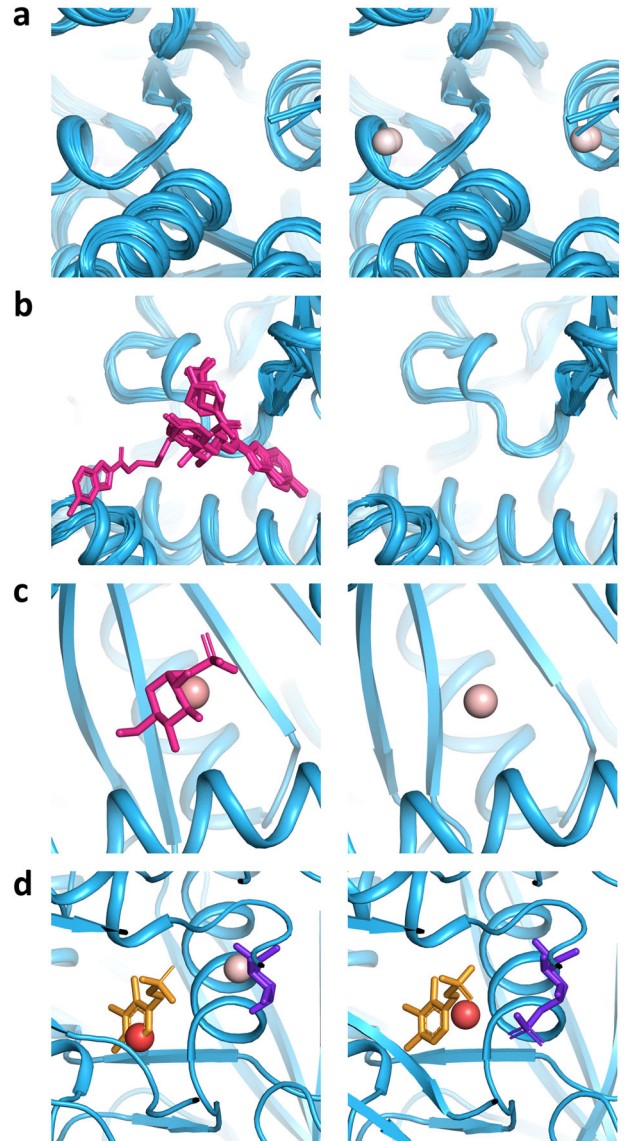

**Fig. 8 Examples of BiteNet prediction errors on the Glycosil transferase protein family (IPR000811). a** Low scored false positive predictions, for which there are no bound ligands. **b** False negative predictions, that is absence of predictions in the proximity of the bound ligand. **c** Similar predictions may correspond to both the true positive and false positive predictions depending on the presence (PDB ID: 3GPB) or absence (PDB ID: 1FU7) of the bound ligand. **d** Catalytic site with two ligands is predicted either as two (PDB ID: 1LWN) or single (PDB ID: 5GPB) binding sites, resulting in false negative predictions. BiteNet predictions are depicted with spheres colored from white to red with respect to the probability score, and ligands are depicted with magenta, yellow and purple sticks.

augmentation and provided random orientation of proteins to the neural network each epoch.

Hyperparameters, such as neural network architecture, type of the activation functions, the learning rate, and many others, influences the model performance. Thus, fine-tuning is needed in order to find optimal set of the hyperparameters. We trained several models and found the following hyperparameters to be optimal: 64 voxels for the cubic grid size, 1.0 Å for the voxel size, 4.0 Å for the density cutoff, 48 for the stride parameter, 16 for the minibatch size, 1e−5 and 10.0 for the $\gamma$ and $\lambda$ parameters, respectively (see Supplementary Table 2 for evaluation of models corresponding to different parameters). Among these parameters,

the voxel size has dramatic influence on the computational speed, it takes ~2 times more to train and apply the model with the voxel size of 0.8 Å, as compared to the voxel size of 1.0 Å. On the other hand, we observed model corresponding to the voxel size of 2.0 Å to be faster, though less accurate. Although we achieved satisfied performance of the resulting model (the average precision was improved from 0.4 to 0.53), our parameter screen is not meticulous. The auto-ml approaches would be useful to find optimal model through extensive search of neural network architecture and parameters[43,44].

Note, that the obtained model is not rotation-translation invariant by construction; it could be easily seen from the different binding site scores assigned to identical subunits of oligomer (see Supplementary Fig. 4). To make sure this does not noticeably affect BiteNet's performance, we re-evaluate the average precision on the augmented validation set and test benchmarks, that contains additional 50 replicas of each protein obtained with rotation by $\pi/3$, $2\pi/3$, $\pi$, $4\pi/3$, and $5\pi/3$ angles about ten different axes corresponding to the centroids of the icosahedron facets[45]. Indeed, we observed that the average precision either did not change or slightly increased (see Supplementary Tables 3–12). This is because for some replicas BiteNet produced additional binding site predictions with very low scores. Next, we analyzed if using of additional rotations affect internal ranking of true binding sites with respect to the probability score for each single structure in the HOLO4K benchmark. We observed that for most of the protein structures (2676 out of 3203) the ranking of the true positive predictions did not change, for 333 protein structures the ranking was improved, and for 194 protein structures the ranking was worsened. Thus, it might be useful to apply BiteNet for different orientations of the structure and average the obtained results.

As the performance metric we used the average precision (AP), that is the area below precision-recall curve:

$$AP = \int_0^1 p(r)\mathrm{d}r,$$

where $p$ is precision (Eq. 4), and $r$ is recall (Eq. 5). The AP metric is one of the most indicative metric for the object detection problems in computer vision used by different methods and benchmarks[46–50]. Note that $p$ or $r$ metrics itself are not indicative, because precision and recall tends to be higher with smaller and larger number of predicted binding sites, respectively. The AP metric, in turn, is independent from the number of predicted binding sites and strongly depends on the ranking of the predictions, thus, it is suitable for comparison of methods with different average number of predicted binding sites and score ranges. We would also like to note that other conventional metrics, like specificity or Matthews correlation coefficient (MCC), are not suitable for comparison due to the lack of strict definition of a true negative (TN) prediction. Indeed, there are literally infinite number of points around the protein structure that can be considered as negative predictions of the binding site centers. Furthermore, as various methods operates with binding sites differently, e.g., voxel centers (BiteNet), surface points (Fpocket and P2Rank), low energy point clusters (SiteHound), it is difficult to rigorously define true negative prediction, that would be suitable for general comparison.

In this study we introduced BiteNet, a deep learning approach for spatiotemporal identification of binding sites. BiteNet takes advantages of the computer vision methods for object detection, by representing three-dimensional structure of a protein as a 3D image with channels corresponding to the atomic densities. BiteNet goes beyond classical problem of binding site prediction in *holo* protein structures, exploring protein dynamics and

flexibility by means of large-scale analysis of conformational ensembles. It is able to detect allosteric binding sites for both soluble and transmembrane protein domains and outperforms state-of-the-art methods both in terms of accuracy and speed. BiteNet takes approximately 0.1 seconds to analyze single conformation and 1.5 minutes to analyze molecular dynamics trajectory with 1000 frames for protein with ~2000 atoms.

## Methods

**Training dataset.** To compose the training set we retrieved atomic structures of protein-ligand complexes with resolution better than 3.0 Å, that contain less than four protein chains, and the sequence identity threshold of 90% from protein data bank (PDB)[51]. Then we refined each protein structure by replacing nonstandard amino acid residues with the standard ones, modeling missing residues and short loops (less than ten amino acid residues) using the ICM-Pro software (molsoft.com). Note, that we did not model N-terminus and C-terminus, as well as long missing loops of more than ten amino acid residues. Then we discarded proteins, if refinement affects three or more atoms of its binding sites, because such conformational changes could be incompatible with the ligand binding pose. We also discarded water molecules, ions, protein chains with length less than 50 amino acid residues, and considered only nondetergent molecules (see Supplementary Table 1) with more than 14 heavy atoms as the ligands. We further disregarded protein complexes with less than 20 protein heavy atoms in the binding site, that is protein atoms within 4 Å distance from the ligand. Finally, we manually filtered out "long" proteins, which length across at least one of the principal axis was more than 250 Å (see Supplementary Fig. 5). This procedure yielded the final set of 5946 atomic structures of protein–ligand complexes comprising 11,301 polypeptide chains and 11,949 binding sites.

We considered each protein of a protein complex as a voxel grid, with voxel size of 1.0 Å with no spacing between the voxels. We represented each voxel by 11 channels corresponding to the atomic density function of a certain atom type, similarly to[52]:

$$\rho(r) = \begin{cases} e^{-r^2/2}, & \text{if } r \le r_{\text{cutoff}} \\ 0, & \text{otherwise} \end{cases}, \tag{1}$$

where $r_{\text{cutoff}}$ is the distance threshold of 4 Å.

For rigorous validation of the prediction model it is important to carefully split the training and the validation datasets. Given that proteins with low sequence similarity may still have high structural similarity, the standard random split would likely lead to the biased training and validation sets. To reduce possible bias, we calculated structural similarity for each pair of protein chains in the dataset using the TMalign software[42], resulting in 11,301 × 11,301 structural similarity matrix (see Supplementary Fig. 6). Then we grouped protein chains using the hierarchical clustering algorithm implemented in *sklearn*[53,54], such that structural similarity of any two protein chains from different clusters is less than 0.5. Finally, we split the dataset in a way that the training and the validation sets do not share protein chains from the same clusters, comprising 9844 and 1457 protein chains, respectively.

**Benchmarks.** The HOLO4K benchmark is a large dataset of holo protein structures used for evaluation of binding site prediction methods[55]; it counts 4542 proteins, most of which are multichain complexes. The original COACH benchmark consists of 501 single chain proteins[56]; in this study, we used the subset of 420 proteins on which several state-of-the-art binding site prediction methods were compared recently[28].

We compared BiteNet with the following approaches for the binding site detection: fpocket, a geometry-based method[23]; SiteHound, that uses probe molecules to find low energy clusters corresponding to the binding sites[21]; MetaPocket, a consensus approach that combines predictions of other methods[24]; DeepSite, a deep learning approach based on the voxelized representation of protein structures[30]; and P2Rank, a classical machine learning approach based on the feature vectors calculated from the protein surface[28].

For fair comparison we considered only proteins not presented in the method's train sets, and for which all methods successfully predict true binding sites according to the P2Rank criterion[28], resulting in the 239 and 1682 protein subsets from COACH420 and HOLO4K, respectively. Also to compute performances of the methods we used both our and P2Rank's definition of the binding site. More precisely, the P2Rank's definition filters small molecules with less than 4 atoms, as well as HOH, DOD, WAT, NAG, MAN, UNK, GLC, ADA, MPD, GOL, SO4, and PO4 molecules. The ligand must be within 4Å of a protein, and the distance from the ligand center to protein must be at least 5.5 Å. The average number of ligand binding sites per protein structure for both criteria for the COACH420 and HOLO4K benchmarks, as well as the average number of predictions of each method are listed in Supplementary Table 13. In addition performances on the entire datasets are provided in Supplementary Tables 5, 6.

**Neural network architecture.** Given $N_x \times N_y \times N_z \times N_c$ voxel grid representation of a protein, we first divided it into the cubic grids of the fixed shape of $64 \times 64 \times 64$ voxels with stride of 48 voxels, in order to get constant size input for the neural network. We considered cubic grids with the average atom density less than 1e−4 as empty cubic grids, and discarded it from the training and validation sets. Following the Yolo approach for the object detection problem in images[50], we constructed neural network that converts $64 \times 64 \times 64$ cubic grid into $8 \times 8 \times 8$ cubic cells of size $8 \times 8 \times 8$ voxels each, and aims to identify target cells, that contain centers of the binding sites, along with the center's coordinates. Thus, the output of the prediction model is $8 \times 8 \times 8 \times 4$ tensor, where the first three dimensions are the cell coordinates with respect to the cubic grid ($i_{\text{cell}}, j_{\text{cell}}, k_{\text{cell}}$), and the four scalars of the fourth dimension are the probability score $\hat{s}$, that the corresponding cell contains center of a binding site, and the coordinates of this center with respect to the cell $\hat{x}, \hat{y}, \hat{z}$. The core of the neural network comprises ten 3D convolutional layers: $\text{Conv3D}_{32} \Rightarrow \text{Conv3D}_{32}^{\text{pool}} \Rightarrow \text{Conv3D}_{32} \Rightarrow \text{Conv3D}_{32} \Rightarrow \text{Conv3D}_{32}^{\text{pool}} \Rightarrow \text{Conv3D}_{64} \Rightarrow \text{Conv3D}_{64} \Rightarrow \text{Conv3D}_{64}^{\text{pool}} \Rightarrow \text{Conv3D}_{128} \Rightarrow \text{Conv3D}_4$, where the subscript number denotes the number of filters. We used kernels of size (3, 3, 3) for each layer, stride of 2 for the pooling layers, and the batch normalization and the rectified linear unit (ReLu) activation function for all layers, except for the last one. Finally, we use the sigmoid activation function to obtain probability score $\hat{s}$ in the range of (0, 1) and relative coordinates $\hat{x}, \hat{y}, \hat{z}$ of the predicted center of the binding site with respect to the cell. The Cartesian coordinates are then calculated according to (Eq. 2):

$$\begin{aligned} \hat{X} &= c_{\text{size}}^x \cdot v_{\text{size}}^x \cdot (i_{\text{cell}} + \hat{x}) + O_x \\ \hat{Y} &= c_{\text{size}}^y \cdot v_{\text{size}}^y \cdot (j_{\text{cell}} + \hat{y}) + O_y , \\ \hat{Z} &= c_{\text{size}}^z \cdot v_{\text{size}}^z \cdot (k_{\text{cell}} + \hat{z}) + O_z \end{aligned} \tag{2}$$

where $c_{\text{size}}$ and $v_{\text{size}}$ corresponds to the size of a cell and voxel, respectively, and $O_x, O_y, O_z$ are the Cartesian coordinates of the origin of the cubic grid.

We used custom loss function for training, that contains three terms:

$$\text{Loss} = \sum_{i=1}^{N_{\text{cells}}} (s_i - \hat{s}_i)^2 + \lambda \sum_{i=1}^{N_{\text{cells}}} s_i \cdot \left( (x_i - \hat{x}_i)^2 + (y_i - \hat{y}_i)^2 + (z_i - \hat{z}_i)^2 \right) + \gamma L_2 , \tag{3}$$

where $N_{\text{cells}}$ is the number of cells in the single cubic grid, $s_i$ and $\hat{s}_i$ are the true (0 or 1) and predicted probability scores of the cell, $x_i, y_i, z_i$ and $\hat{x}_i, \hat{y}_i, \hat{z}_i$ are the true and predicted coordinates for $i$th cell, respectively, and $L_2$ correspond to the regularization term. Therefore, the first and the second terms aim to penalize incorrect prediction of the probability score and the center of the binding site, respectively. Note, that we multiply the second term by the true probability score (0 or 1) to take into account only relevant predictions. The third term is the $L_2$ regularization term for the neural network parameters. The coefficients $\lambda = 5$ and $\gamma = 1e−5$ are the weights of the penalty terms.

We trained the network in Tensorflow v1.14[57] for 400 epochs using the Adam optimizer with the default parameters, minibatch size of 16 cubic grids, and the learning rate of 1e−3 gradually decreasing to 1e−5 during the training. We would like to note, that presented architecture is not invariant to rotations of a protein. Data augmentation, i.e., considering different orientations of a protein within a single epoch, may circumvent this problem to some extent. Because of GPU memory limitations, in this study, we used implicit data augmentation by considering random orientation of a protein each epoch.

To obtain the final predictions we applied the post-processing procedure, as it follows. First, we discarded all the predictions with the probability score $\hat{s} < s_{\text{threshold}}$. The remaining predictions are then processed by means of the non-maximum suppression. More precisely, we select the best prediction in terms of the probability score, as the seed of a cluster, and put all the predictions with the centers of the binding site closer than $d_{\text{threshold}} = 8$ Å to the center of the best prediction. Then we select the second best prediction, as the seed of the next cluster, and repeat the above procedure until all the predictions are clustered. Finally, we keep only $N_{\text{top}}$ seeds in terms of the probability scores, as the final predictions. For the training we used $s_{\text{threshold}} = 0.1$ and $N_{\text{top}} = 5$, for benchmarking $s_{\text{threshold}} = 0.01$ (in order to calculate AP for all predictions), and for the case study $s_{\text{threshold}} = 0.1$ and all predictions. To evaluate the performance of the prediction model we define the true positive (TP) prediction of the binding site, as the top-scored correct prediction, that is prediction with the probability score $\hat{s} \ge s_{\text{threshold}}$ and the predicted center of the binding site within $d_{\text{threshold}}$ from the true center of the binding site. The rest of the predictions are considered as false positives (FP). Given this, we calculate precision and recall metrics according to:

$$\text{Precision} = \frac{N_{\text{TP}}}{N_{\text{TP}} + N_{\text{FP}}}, \tag{4}$$

$$\text{Recall} = \frac{N_{\text{TP}}}{N_{\text{TP}} + N_{\text{FN}}}, \tag{5}$$

where $N_{\text{FN}}$ is the number of false negative predictions, that is the number of binding sites with no correct prediction. As the main metric we calculate the average precision metric AP, which is the area under precision recall curve.

Note, that we define correct prediction with respect to the center of the binding site, rather than binding pose of a ligand. We believe this is more rigorous metric,

because it does not depend neither on the binding pose of a ligand, nor on the ligand itself. However, for fair comparison with the existing methods, we also computed the metrics, where the prediction is considered to be true positive prediction, if the minimal distance to the ligand is less than $d_{threshold} = 4Å$.

**Clusterization**. Given conformational ensemble of a protein, as for example, molecular dynamics trajectory, we firstly applied BiteNet to each conformation. Then we grouped the obtained predictions using clustering algorithms. In this study, we used three different clustering approaches implemented in the sklearn python library[54]: the mean shift clustering algorithm (MSCA)[58], the density-based clustering algorithm (DBSCAN)[59,60], and the agglomerative hierachical clustering algorithm[53]. While the first two approaches are mainly applied for the set of points in Euclidean space, the latter approach can be applied also for set of amino acid residues forming the predicted binding site. Finally, we assigned two scores for each cluster. The first score is the sum of maximal probability score of a cluster in each frame averaged over the total number of frames. For the second score, the mean sum of probabilities scores (larger than $s_{cluster\_score\_threshold\_step} = 0.1$) of a cluster is computed for each frame; these sums are then averaged over the total number of the corresponding frames. We implement several clustering approaches, because it is known that clustering results may strongly vary depending on clustering algorithm and different parameters for them, also affecting the cluster scores.

**Statistics and reproducibility**. To support significant outperforming of BiteNet over the other methods, we performed statistical Student's test, as it follows. We considered protein structures from the COACH420 and HOLO4K benchmarks that are not in the BiteNet's training set and have at least one binding site according to the P2Rank filtering criterion. Then we split these protein structures into 31 independent subsets, and evaluated the performance of each method for each subset. Finally, we considered the null hypothesis that there is no significant difference in performance metrics between BiteNet and other methods. The highest calculated $p$-value for the AP metric is $1.2e^{-6}$, that allows us to reject the null hypothesis (see Supplementary Table 14 for more details). The BiteNet model required to reproduce the results of this study are available at https://github.com/i-Molecule/bitenet and https://doi.org/10.5281/zenodo.4043664[61]. In addition a web-server implementation of BiteNet is available at https://sites.skoltech.ru/imolecule/tools/bitenet.

**Reporting summary**. Further information on research design is available in the Nature Research Reporting Summary linked to this article.

## Data availability
The dataset used for training of BiteNet is available at https://doi.org/10.5281/zenodo.4043664[61]. BiteNet is available at https://github.com/i-Molecule/bitenet.

## Code availability
BiteNet source code is available at https://github.com/i-Molecule/bitenet and https://doi.org/10.5281/zenodo.4043664[61].

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

## Acknowledgements

We acknowledge the HPC team at CDISE (Skoltech) for support usage of the "Zhores" supercomputer in order to train BiteNet.

## Author contributions
I.K. and P.P. constructed the training, validation, and test sets, processed protein structures, formulated the machine learning problem, developed BiteNet, conducted numerical experiments, performed data analysis and wrote the manuscript. P.P. organized and managed the project implementation, and supervised the research.

## Competing interests
The authors declare no competing interests.
