## [Peer Review File · Communications Biology]

Reviewers' comments:

Reviewer #1 (Remarks to the Author):

In this study, the authors have proposed a deep-learning-based approach to identify the binding pocket from the given 3D structure. BiteNet considers protein conformations as the 3D images and the binding sites as objects on these images to detect. They evaluated their methods on different independent datasets and the results show that BiteNet significantly outperformed the existing methods. Although this article is interesting, it cannot attract a general audience. It is more suitable for the Journal of chemical information and modeling or Bioinformatics. I have the following comments to help improve the presentation of the manuscript:

The current evaluation of BiteNet using COACH420 and HOLO4K seems biased. Exclude similar protein (COACJ420 and HOLO4K) from the training dataset before the prediction model construction. And use the entire dataset for the evaluation, and compare it with the existing methods.

While constructing a training dataset, a lower sequence identity will be preferred to avoid overestimation and robustness. However, the authors employed 90% cut-off, which is too high. It is better to try different cut-offs and show the optimal one.

Conventionally, k-fold cross-validation (CV) or leave-one-out cross-validation is used to develop a prediction model. It is not clear what kind of CV is used in this study.

Under spatiotemporal prediction of binding sites in pharmacological targets, three examples (ATP-gated ion channel, EGFR, and GPCR) are given. I am wondering how much the sequence identity of these three proteins shares with their training dataset. If they have a similar structure, this performance is quite obvious. Why they are applying a different protocol? Specifically, they generated MD trajectory on two cases and neglected in one case. When you have an experimental result, it is a quite obvious one can apply the different procedure and obtain the result closer to the experimental data. I cannot see any systematic protocol.

I suggest the authors provide a web-server rather than a standalone version. Web-server will be more helpful to the experimentalists.

Reviewer #2 (Remarks to the Author):

Kozlovskii and Popov present an interesting study that leverages deep learning to identify druggable binding sites, by combining 3D convolutional neural networks with protein dynamics. The study is of great interest to the scientific community, also thanks to the code availability.

However, in my opinion, there are several aspects to be considered to improve the manuscript clarity and transparency, see my comments below:

Major

-The model presented by the authors is not rotation invariant. The authors provide a small visual aid, saying that: "To ensure, that this is not an artefact of the rotational variance of the model, we generate 50 replicas by rotating the monomer about 10 axes by $\pi/3$, $2\pi/3$, π , $4\pi/3$, and $5\pi/3$ angles and averaged the obtained predictions. As one can see from Figure 2D, although the absolute values of the probability scores vary with respect to the monomers, in all the cases BiteNet correctly identifies the allosteric binding site for the trimer complex and not for the monomer":

1. In my opinion, this visual aid is not systematic enough to really check for the effect of the rotation. Some more analysis on how this may vary depending on the rotation of the protein has

to be included, to really check for the importance of this “external” elements and guide the reader/potential user.

2. Also, it is not clear how a hypothetical user should utilize the model. Shall one perform several rotations and check for the consistency?

3. Are the performance metrics provided obtained by considering multiple rotations or just one?
- Model validation and performance quantification.

1. The authors use recall and precision to quantify the model performance; both metrics mainly quantify the ability to detect true positives. However, I believe that specificity (which quantifies the ability of not generating false positives, see Chemometrics and Intelligent Laboratory Systems 174 (2018): 33-44 for the formula) in prospective application could be equally (if not even more important) than precision and recall. In fact, often the task is to identify with certainty at least one druggable binding site. Please, add specificity to the pool of metrics utilized for performance quantification.

2. Despite precision is often used, it does not consider the different numerosity in the “P” and “N” classes and it is, thus, in my opinion not a good metric to compare among different proteins with different numbers of binding sites. Please also provide recall and specificity metrics (e.g., Fig. 5) as a way to quantify the goodness of the predictions.

o The authors exclude the proteins for which all the methods successfully predicted the binding sites. Despite I get the point of this, removing these samples will somehow distort the performance quantified and emphasize differences that are potentially not significant. I recommend preserving this information (i.e., the performance quantified on the whole set) somewhere, e.g., in the supporting information.

3. Please perform a statistical test to support the claim of outperforming the other methods, on all the metrics mentioned.

Minor

Introduction

- Clarify the meaning of “conformational ensembles”
- Replace “the most difficult binding site”, as it is something not really measurable

Results

- “To demonstrate...coupled receptor family”. What are the scientific bases to state that these targets are difficult? Please, insert some references to literature.
- Please, specify how the benchmark methods were chosen.
- Fig. 5: instead of reporting average metrics in an error bar, please report boxplots with all the “raw” values, so that the underlying information is clearer.

Reviewer #3 (Remarks to the Author):

This manuscript presents BiteNet, 3D convolutional neural network model to predict the location, extension and probability of a binding site in a crystal structure. Manually curated protein structures from the Protein Data Bank were used as the training set and several test sets (including benchmarks COACH420 and HOLO4K). The authors claim the superiority of their method based on average precision (figure 5A).

The most important part of the paper is this comparison to other methods on COACH420 and HOLO4K, which unfortunately was poorly made. The entire comparison is carried in two short paragraphs in section C. The following information is needed for a proper comparison:

- 1) a minimal description of the benchmark is required for the study to be reasonably self-contained: e.g. how many of those proteins have 4542 proteins have a known binding site? How many binding sites per protein on average? How were those binding sites defined? And their coordinates?
- 2) Figure 5A shows the average precision for each method and benchmark. It is not clear what this average refers to. There are N known binding sites, a single number of TP and FP for each method,

therefore a single precision value per method.

3) To show the global performance of these methods, Figure 5A should report the Matthews Correlation Coefficient (MCC) of each method instead.

4) In addition to the comparison of MCCs on all proteins, it is important to show the same comparison on each major protein class. Does the ranking of the methods change depending on the protein class?

5) Looking at the FPs of BiteNet, what types of proteins have binding sites hard to detect?

6) Looking at the FNs of BiteNet, what types of proteins seem to have binding sites but do not?

7) BiteNet is compared to other methods on a part of COACH420 and HOLO4K proteins to avoid overlap with training set. I still would like to see the comparison in the full test set (in the supplementary materials).

8) The authors said "We trained several models and found the following hyperparameters to be optimal". I found this worrying: where they optimal because they provide the best precision on the test sets? If so, BiteNet performance on those sets would be overestimated (an unfair comparison).

To make run for the new experiments, some/all the case studies where only BiteNet is employed can be moved to the supplementary materials.

We are grateful to the reviewers for their insightful comments and suggestions that help us to improve the manuscript. We substantially revised the manuscript and addressed the reviewer's comments point-by-point with the indicated changes in the manuscript below.

As several concerns were raised regarding model validation and performance quantification, we would like to clarify the used performance metric, as it seems that it was not properly described in the former version of the manuscript. First of all we completely agree with the reviewers that fair and rigorous comparison with respect to the state-of-art method is a must, and we did calculate the conventional performance metrics for comparison, such as precision, recall, specificity, and MCC. However, these metrics either can be easily overtrained, or involve the TN term, that has no rigorous definition for the binding site prediction problem, as there are literally infinite number of points around a protein structure that can be classified as "not a binding site point". Moreover, as various methods operates with binding sites differently, e.g. voxel centers (BiteNet), surface points (fpocket and P2Rank), low energy point clusters (SiteHound), it is difficult to define true negative prediction, that would be suitable for general comparison. The precision and recall metrics rely on TP, FP, and FN metrics, that are defined for all methods. However, precision and recall tends to be higher with smaller and larger number of predicted binding sites, respectively, as was also pointed out by one of the reviewers. So among all the metrics, we found "The Average Precision" (AP) metric (has nothing in common with the "averaged precision") the most rigorous for the fair comparison. AP is one of the main metrics in object detection problems, it is independent from the number of predicted binding sites and strongly depends on the ranking of the predictions, thus, it is suitable for comparison of methods with different average number of predicted binding sites and score ranges.

To clarify these we introduced the following changes into the manuscript:

p. 10 added section Metrics

As the performance metric we used the average precision (AP), that is the area below precision-recall curve:

Eq.

where p is precision (Eq. 4), and r is recall (Eq. 5). The AP metric is one of the most indicative metric for the object detection problems in computer vision used by different methods and benchmarks [46–50]. Note that p or r metrics itself are not indicative, because precision and recall tends to be higher with smaller and larger number of predicted binding sites, respectively. The AP metric, in turn, is independent from the number of predicted

binding sites and strongly depends on the ranking of the predictions, thus, it is suitable for comparison of methods with different average number of predicted binding sites and score ranges. We would also like to note that other conventional metrics, like specificity or Matthews Correlation Coefficient (MCC), are not suitable for comparison due to the lack of strict definition of a true negative (TN) prediction. Indeed, there are literally infinite number of points around the protein structure that can be considered as negative predictions of the binding site centers. Furthermore, as various methods operate with binding sites differently, e.g. voxel centers (BiteNet), surface points (Fpocket and P2Rank), low energy point clusters (SiteHound), it is difficult to rigorously define true negative prediction, that would be suitable for general comparison.

Reviewer 1

The current evaluation of BiteNet using COACH420 and HOLO4K seems biased. Exclude similar protein (COACH420 and HOLO4K) from the training dataset before the prediction model construction. And use the entire dataset for the evaluation, and compare it with the existing methods.

Answer:

We separately trained predictive models on two reduced datasets with excluded structures from the COACH420 and HOLO4K benchmarks, respectively. The datasets contain 5155/726 and 4856/686 structures in the train/validation sets, respectively, as compared to the initial 5212/734 dataset. We trained both models using the same parameters as in our final BiteNet model and evaluated its performance on the COACH420 and HOLO4K benchmarks. We did not observe any bias, as there is no much difference in the AP metrics:

Benchmark	AP for BiteNet trained on initial dataset	AP for BiteNet trained on initial dataset without COACH420 structures	AP for BiteNet trained on initial dataset without HOLO4K structures
COACH420	0.68/0.58*	0.68/0.57	0.69/0.59
HOLO4K	0.77/0.66	0.78/0.67	0.77/0.66

* first and second numbers correspond to the P2Rank-based and BiteNet-based definition of binding sites, respectively

Changes in manuscript:

Supplementary Information: page 12; Table 11: Performance metrics for the BiteNet model (default), and models trained on datasets with excluded proteins from the COACH420 (wo_coach) and HOLO4K (wo_holo) benchmarks. These datasets contain 5155/726 and 4856/686 structures in the training/validation sets, respectively.

While constructing a training dataset, a lower sequence identity will be preferred to avoid overestimation and robustness. However, the authors employed 90% cut-off, which is too high. It is better to try different cut-offs and show the optimal one.

Answer:

We disagree with the reviewer, as there are protein structures with different binding sites, but similar (or even identical) sequences. Therefore, Introducing a lower sequence identity threshold (SI) will result in keeping only one of such structures, hence, information loss, rather than increase in robustness. Furthermore, as the dataset size strongly depends on SI, the comparison of models corresponding to the different SIs is not straightforward. We believe that taking into account structural similarity for the train/validation split is much more important to derive a robust model. This is why we used 50% of structural similarity threshold, rather than SI, in order to separate dissimilar proteins to the training and validation sets. Thus, for any pair of proteins from training and validation the structure similarity is lower than 50%.

Conventionally, k-fold cross-validation (CV) or leave-one-out cross-validation is used to develop a prediction model. It is not clear what kind of CV is used in this study.

Answer:

We use conventional train-validation split to derive the predictive model, rather than K-fold CV, as K-fold CV will result in similar protein structures presented in train and validation folds, hence, bias towards the most represented protein family.

Changes in the manuscript:

We changed ‘test set’ to ‘validation set’ throughout the manuscript to be consistent with the terminology.

Under spatiotemporal prediction of binding sites in pharmacological targets, three examples (ATP-gated ion channel, EGFR, and GPCR) are given. I am wondering how much the sequence identity of these three proteins shares with their training dataset. If they have a similar structure, this performance is quite obvious. Why they are applying a different protocol? Specifically, they generated MD trajectory on two cases and neglected in one case. When you have an experimental result, it is a quite obvious one can apply the different procedure and obtain the result closer to the experimental data. I cannot see any systematic protocol.

Answer:

We agree that binding sites of the case studies should not overlap with the binding sites present in the training set, and this is how we selected our case studies in the first place. So there are no similar binding sites in the training set. More specifically, the maximum sequence identity between the P2X3 protein (PDB IDs: 5SVK, 5YVE) and the training set is 0.32 (PDB ID: 5L9Z), structural similarity is 0.60 (PDB ID: 6J4P, transmembrane region with no binding site). For the next case study, the training set does contain two EGFR kinase domain structures (PDB IDs: 5UG9, 5GNK) as well as three other proteins with high sequence identity and structure similarity: DDX25 RNA helicase (PDB ID: 2RB4), kinase domain of human HER2 (erbB2) (PDB ID: 3PP0) and HER3 pseudokinase domain (PDB ID: 4OTW) with sequence similarity of 0.72, 0.76 and 0.60 and structure similarity 0.83, 0.88 and 0.94, respectively. Although these structures possess similar *orthosteric* binding sites, none of these structures possess the *allosteric* binding site, which is the main focus of this case study. As for the A2A receptor, we demonstrate a *hypothetical allosteric* binding site detected by BiteNet, that is not present in the training set. Therefore the presented case studies are not a simple showcase of the memorized samples from the training set.

In all case studies we applied the same BiteNet model (the same protocol) to different molecular systems in order to show possible application of BiteNet. The first case study is to show application on multimeric protein complexes, the second is to show conformational specificity of our model, and the third is to show possible ways to investigate novel binding sites based on the MD trajectories using BiteNet. As for the benchmarks studies we also applied the same BiteNet model and did not tweak any parameter to achieve better performance or fit better the test benchmarks.

Changes in the manuscript:

Main text: page 2; Section II-B-1:

We would like to emphasize that the training set does not contain structures similar to the P2X3 receptor. Indeed, the maximal sequence identity is 0.32 for human heparanase (PDB ID: 5L9Z) and the maximal structure similarity is 0.6 for tyrosine carboxypeptidase (PDB ID: 6J4P). Thus, this case demonstrates predictive power of BiteNet, rather than detection of memorized binding sites.

Main text: page 3; Section II-B-2:

In contrast to the P2X3 case study, the training set does contain two EGFR kinase domain structures (PDB IDs: 5UG9, 5GNK) as well as three other proteins with high sequence identity and structure similarity: DDX25 RNA helicase (PDB ID: 2RB4), kinase domain of human HER2 (erbB2)(PDB ID: 3PP0) and HER3 pseudokinase domain (PDB ID: 4OTW) with sequence similarity of 0.722, 0.762 and 0.596 and structure similarity 0.833, 0.876 and 0.944, respectively. Nonetheless, all these structures have ligands bound to the binding sites corresponding to the EGFR orthosteric binding site, but not to the allosteric binding site. Therefore, this example shows the predictive power of BiteNet to detect conformation-specific binding sites.

I suggest the authors provide a web-server rather than a standalone version. Web-server will be more helpful to the experimentalists.

Answer:

We thank the reviewer for this suggestion and we implemented BiteNet on a local server. It is now available at <https://sites.skoltech.ru/imolecule/tools/bitenet>

Changes in the manuscript:

Added to Conclusions (page 13):

and as a web-server application at <https://sites.skoltech.ru/imolecule/tools/bitenet>.

Reviewer 2

-The model presented by the authors is not rotation invariant. The authors provide a small visual aid, saying that: "To ensure, that this is not an artefact of the rotational variance of the model, we generate 50 replicas by rotating the monomer about 10 axes by $\pi/3$, $2\pi/3$, π , $4\pi/3$, and $5\pi/3$ angles and averaged the obtained predictions. As one can see from Figure 2D, although the absolute values of the probability scores vary with respect to the monomers, in all the cases BiteNet correctly identifies the allosteric binding site for the trimer complex and not for the monomer":

1. In my opinion, this visual aid is not systematic enough to really check for the effect of the rotation. Some more analysis on how this may vary depending on the rotation of the protein has to be included, to really check for the importance of this “external” elements and guide the reader/potential user.

Answer:

For more systematic assessment of the effect of rotations we evaluated the BiteNet model on the training set, validation set and the benchmarks augmented with the 50 replicas corresponding to the rotations along the 10 different axes and 5 angles. We observed either none or only minor improvement in AP, as compared to the sets and benchmarks with no replicas. In addition we analyzed the internal ranking of true binding sites based with respect to the probability score for each structure in the HOLO4K dataset. We observed that for ~84% of protein structures (2676 out of 3203) the ranking of the true positive predictions did not change, for ~10% (333) of protein structures the ranking was improved, and for ~6% (194) of protein structures the ranking was worsened.

Changes in the manuscript:

Supplementary Information: page 2-11; Supplementary Tables 1-10:

Added BiteNet R+ model, for which predictions are calculated taking into account 50 additional rotations.

Supplementary Information: page 13; Supplementary Table 12:

Comparison of BiteNet and BiteNet R+ models on the training and validation sets.

Added to Section III-B (page 10):

Indeed, we observed that the average precision either did not change or slightly increased (see Supplementary Tables 1-10). This is because for some replicas BiteNet produced additional binding site predictions with very low scores. Next we analyzed if using of additional rotations affect internal ranking of true binding sites with respect to the probability score for each single structure in the HOLO4K benchmark. We observed that for most of the protein structures (2676 out of 3203) the ranking of the true positive predictions did not change, for 333 protein structures the ranking was improved, and for 194 protein structures the ranking was worsened.

2. Also, it is not clear how a hypothetical user should utilize the model. Shall one perform several rotations and check for the consistency?

Answer:

We suggest to use additional rotations whenever the elapsed computational time remains feasible, which is the case of multiple protein structures or conformations, as well as small number of trajectory frames (for example analysis of 1000 frames with additional rotations takes about one hour). However we did not observe much differences in performance of BiteNet applied to protein structures with or without additional rotations (see the previous answer). For the user convenience we added an option to include additional rotations in the web-server implementation of BiteNet.

3. Are the performance metrics provided obtained by considering multiple rotations or just one?

Answer:

In the main text we list the performance metrics for the predictions obtained without additional rotations.

-Model validation and performance quantification.

*1. The authors use recall and precision to quantify the model performance; both metrics mainly quantify the ability to detect true positives. However, I believe that specificity (which quantifies the ability of not generating false positives, see *Chemometrics and Intelligent Laboratory Systems* 174 (2018): 33-44 for the formula) in prospective application could be equally (if not even more important) than precision and recall. In fact, often the task is to identify with certainty at least one druggable binding site. Please, add specificity to the pool of metrics utilized for performance quantification.*

2. Despite precision is often used, it does not consider the different numerosity in the “P” and “N” classes and it is, thus, in my opinion not a good metric to compare among different proteins with different numbers of binding sites. Please also provide recall and specificity metrics (e.g., Fig. 5) as a way to quantify the goodness of the predictions.

Answer:

We agree with the reviewer, that the ability of not generating false positives is important, and this is why we list the AP metric, rather than precision and recall in the main text. As for the specificity, note that there is no strict definition of the TN metric required to compute it (please see our first comment). For example, to define TN for our model one could consider all predictions obtained from each cell ($8*8*8=512$ predictions), classify them as T or N w.r.t

the probability score threshold of 0.01, and discriminate TP based on the distance threshold to the binding site of 4Å. Thus, TN corresponds to the predictions with the probability score ≤ 0.01 and distance to any true binding site $\geq 4\text{Å}$. The specificity for such a scenario calculated for BiteNet varies from 0.98 to 0.99 on COACH420 and HOLO4K benchmarks, depending on the definition of the binding site. However such specificity cannot be calculated, for example, for fpocket or P2Rank predictions, so we prefer not to include this information into the manuscript, as it could be confusing or misleading to a reader. Instead we listed FP, TP, and FN metrics for all the methods in the supplementary materials.

Changes in the manuscript:

Supplementary Information: pages 2-11: Supplementary Tables 1-10:

In addition to the precision, recall and average precision, TP, FP, and FN scores are listed for all the methods.

The authors exclude the proteins for which all the methods successfully predicted the binding sites. Despite I get the point of this, removing these samples will somehow distort the performance quantified and emphasize differences that are potentially not significant. I recommend preserving this information (i.e., the performance quantified on the whole set) somewhere, e.g., in the supporting information.

Answer:

We agree with the reviewer and added comparison of the described methods on i) the entire COACH420 and HOLO4K datasets, and ii) its subsets with excluded structures from BiteNet train or validation sets, comprising proteins with at least one binding site based on a) P2Rank criterion, and b) our criterion. In all the cases we observed the same performance trend and listed the calculated metrics in the supporting information.

Changes in the manuscript:

In Supporting Information:

Table 1: Performance metrics on the COACH420 subset of 239 proteins, which are not in the BiteNet training set, outputs of all methods are available for them and each structure contains at least one relevant binding site according to the P2Rank binding site filtering criterion.

Table 2: Performance metrics on the HOLO4K subset of 1682 proteins, which are not in the BiteNet training set, outputs of all methods are available for them and each structure contains at least one relevant binding site according to the P2Rank binding site filtering criterion.

Table 3: Performance metrics for the entire COACH420 dataset.

Table 4: Performance metrics for the entire HOLO4K dataset.

Table 5: Performance metrics on the COACH420 subset of 352 proteins, which are not in the BiteNet training set and each structure contains at least one relevant binding site according to the P2Rank binding site filtering criterion.

Table 6: Performance metrics on the HOLO4K subset of 4169 proteins, which are not in the BiteNet training set and each structure contains at least one relevant binding site according to the P2Rank binding site filtering criterion.

Table 7: Performance metrics on the COACH420 subset of 344 proteins, which are not in both BiteNet training and validation set and each structure contains at least one relevant binding site according to the P2Rank binding site filtering criterion.

Table 8: Performance metrics on the HOLO4K subset of 4121 proteins, which are not in BiteNet training and validation sets and each structure contains at least one relevant binding site according to the P2Rank binding site filtering criterion.

Table 9: Performance metrics on the COACH420 subset of 238 proteins, which are not in the BiteNet training set and each structure contains at least one relevant binding site according to the P2Rank binding site filtering criterion.

Table 10: Performance metrics on the HOLO4K subset of 2848 proteins, which are not in the BiteNet training set and each structure contains at least one relevant binding site according to the BiteNet binding site filtering criterion.

3. Please perform a statistical test to support the claim of outperforming the other methods, on all the metrics mentioned.

Answer:

We considered the null hypothesis that there is no significant difference in the performance between BiteNet and the other methods. To perform statistical Student's test we considered protein structures from the COACH420 and HOLO4K benchmarks that are not in the BiteNet's training set and have at least one binding site according to the P2Rank filtering

criterion. Then we split these protein structures into 31 independent subsets, and evaluated the performance of each method for each subset. The highest measured p-value corresponding to the AP metric is $1.2e^{-6}$, which allows us to reject the null hypothesis.

Changes in the manuscript:

Main text, page 6:

To compare BiteNet with the other approaches we evaluated its performance on the HOLO4K and COACH420 benchmarks (see Section IV). As the performance metric we used the average precision (AP), as we consider this metric the most suitable for the binding site prediction problem (see Section IIIC). We calculated AP for All and TopN predictions, where N is the number of the true binding sites present in a protein structure. As one can see from Figure 5 BiteNet significantly outperforms ($p\text{-value} \leq 1.2e^{-6}$) classical binding site prediction methods, such as fpocket [23], SiteHound [21], MetaPocket [24], as well as the state-of-the-art machine learning methods, such as DeepSite [30] and P2Rank [28] (Supplementary Tables 1-10 lists more detailed comparison including the performance on the entire benchmarks, as well as the precision, recall, true positive, false positive, and false negative metrics).

Main text, page 13: Statistics and Reproducibility

To support significant outperforming of BiteNet over the other methods, we performed statistical Student's test, as it follows. We considered protein structures from the COACH420 and HOLO4K benchmarks that are not in the BiteNet's training set and have at least one binding site according to the P2Rank filtering criterion. Then we split these protein structures into 31 independent subsets, and evaluated the performance of each method for each subset. Finally, we considered the null hypothesis that there is no significant difference in performance metrics between BiteNet and other methods. The highest calculated p-value for the AP metric is $1.2e^{-6}$, that allows us to reject the null hypothesis (see Supplementary Table 13 for more details). The BiteNet model required to reproduce the results of this study are available at <https://github.com/i-Molecule/bitenet>. In addition a web-server implementation of BiteNet is available at <https://sites.skoltech.ru/imolecule/tools/bitenet>.

Supplementary Information: page 16

Table 13: Statistical tests for comparison of performance of BiteNet with other methods on the set composed from COACH420 and HOLO4K subsets (238 and 1682 proteins, which are not in the BiteNet training set, predictions of all methods available and each structure contains at least one relevant binding site according to the in P2Rank binding site filtering criterion). The T-test is performed: the composed set was split into 31 independent subsets; for each subset precision, recall and average precision (AP) are calculated; the null

hypothesis states that there is no significant difference in performance metrics between BiteNet and another method.

Minor

Introduction

- Clarify the meaning of “conformational ensembles”
- Replace “the most difficult binding site”, as it is something not really measurable

Results

- “To demonstrate...coupled receptor family”. What are the scientific bases to state that these targets are difficult? Please, insert some references to literature.
- Please, specify how the benchmark methods were chosen.
- Fig. 5: instead of reporting average metrics in an error bar, please report boxplots with all the “raw” values, so that the underlying information is clearer.

Answer:

We thank the reviewer for the careful reading of the manuscript and addressed all the minor comments.

Changes in the manuscript:

Main text; page 1; Section I:

Replaced “the entire conformational ensemble”
to “the entire conformational space”

Replaced “conformational ensembles as the 3D videos to analyze”
to “conformational ensembles, that is a set of protein conformations, as the 3D videos to analyze”

Replaced “We showed that BiteNet is capable to solve the most difficult binding site detection challenges”
to “We showed that BiteNet is capable to solve challenging binding site detection problems”

sentence rephrased : Particularly, BiteNet correctly identified oligomer-specific allosteric binding site formed by the subunits of the trimeric P2X3 receptor complex; and conformation-specific allosteric binding site of the epidermal growth factor receptor kinase domain.

Main text; page 2; Section II-B:

Replaced: “one of the most difficult binding site detection challenges”
to “challenging binding site detection problems”

Main text: page 11; Section IV-Benchmarks:

The HOLO4K benchmark is a large dataset of holo protein structures used for evaluation of binding site prediction methods [55]; it counts 4542 proteins, most of which are multichain complexes. The original COACH benchmark consists of 501 single chain proteins [56]; in this study we used the subset of 420 proteins on which several state-of-the-art binding site prediction methods were compared recently [28].

We compared BiteNet with the following approaches for the binding site detection: fpocket, a geometry-based method [23]; SiteHound, that uses probe molecules to find low energy clusters corresponding to the binding sites [21]; MetaPocket, a consensus approach that combines predictions of other methods [24]; DeepSite, a deep learning approach based on the voxelized representation of protein structures [30]; and P2Rank, a classical machine learning approach based on the feature vectors calculated from the protein surface [28].

For fair comparison we considered only proteins not presented in the method’s train sets, and for which all methods successfully predict true binding sites according to the P2Rank criterion [28], resulting in the 239 and 1682 protein subsets from COACH420 and HOLO4K, respectively. Also to compute performances of the methods we used both our and P2Rank’s definition of the binding site. More precisely, the P2Rank’s definition filters small molecules with less than 4 atoms, as well as HOH, DOD, WAT, NAG, MAN, UNK, GLC, ADA, MPD, GOL, SO4, PO4 molecules. The ligand must be within 4Å of a protein, and the distance from the ligand center to protein must be at least 5.5Å. The average number of ligand binding sites per protein structure for both criteria for the COACH420 and HOLO4K benchmarks, as well as the average number of predictions of each method are listed in Supplementary Table 16.

In addition performances on the entire datasets are provided in Supplementary Tables 3,4.

Reviewer 3

1) a minimal description of the benchmark is required for the study to be reasonably self-contained: e.g. how many of those proteins have 4542 proteins have a known binding site? How many binding sites per protein on average? How were those binding sites defined? And their co-ordinates?

Answer:

We included information about the benchmarks into the manuscript. For rigorous comparison we used definition of a binding site as in the P2Rank paper (ligand with ≥ 5 atoms, distance from protein $\leq 4\text{\AA}$, ligand name not in list (HOH, DOD, WAT, NAG, MAN, UNK, GLC, ADA, MPD, GOL, SO₄, PO₄)), as well as our criterion of a binding site used to compose the training and validation sets (ligand with ≥ 14 atoms, protein interface size ≥ 20 atoms, and ligand name not in the the specified list (Supplementary Table 14)). The number of proteins in the benchmarks as well as the average number of binding sites per structure vary w.r.t. the definition. The HOLO4K (COACH420) benchmark contains 4524 and 3203 (409 and 290) proteins with at least one true ligand binding site with the average number of binding sites per structure 2.37 and 1.84 (1.43 and 1.05) binding sites per structure for P2Rank-based and BiteNet-based definition, respectively. The predicted coordinates are considered true positive, if the distance to the ligand atom is lower or equal to 4Å. We calculated all the performance metrics separately for both definitions and listed them in Supplementary information. As the performance metrics also vary w.r.t. the definition of the true positive prediction, we provided particular example of the Aspartic peptidase A1 protein family in the main text.

Changes in the manuscript:

Main text: page 11; Section IV-Benchmarks:

The HOLO4K benchmark is a large dataset of holo protein structures used for evaluation of binding site prediction methods [55]; it counts 4542 proteins, most of which are multichain complexes. The original COACH benchmark consists of 501 single chain proteins [56]; in this study we used the subset of 420 proteins on which several state-of-the-art binding site prediction methods were compared recently [28].

We compared BiteNet with the following approaches for the binding site detection: fpocket, a geometry-based method [23]; SiteHound, that uses probe molecules to find low energy clusters corresponding to the binding sites [21]; MetaPocket, a consensus approach that combines predictions of other methods [24]; DeepSite, a deep learning approach based on the voxelized representation of protein structures [30]; and P2Rank, a classical machine learning approach based on the feature vectors calculated from the protein surface [28].

For fair comparison we considered only proteins not presented in the method's train sets, and for which all methods successfully predict true binding sites according to the P2Rank criterion [28], resulting in the 239 and 1682 protein subsets from COACH420 and HOLO4K, respectively. Also to compute performances of the methods we used both our and P2Rank's definition of the binding site. More precisely, the P2Rank's definition filters small molecules with less than 4 atoms, as well as HOH, DOD, WAT, NAG, MAN, UNK, GLC, ADA, MPD,

GOL, SO4, PO4 molecules. The ligand must be within 4Å of a protein, and the distance from the ligand center to protein must be at least 5.5Å. The average number of ligand binding sites per protein structure for both criteria for the COACH420 and HOLO4K benchmarks, as well as the average number of predictions of each method are listed in Supplementary Table 16.

In addition performances on the entire datasets are provided in Supplementary Tables 3,4.

Main text: page 7

We observed that BiteNet's performance is 5% higher, when the true positive prediction of a binding site is defined as in the training, as compared to the P2Rank's criterion. The main reason for this is more strict ligand filtering implemented in the training aiming to discard not relevant small molecules. For example, in the HOLO4K benchmark there are 29 structures corresponding to the Aspartic peptidase A1 protein family, that have the only active site with a peptide-like molecule bound to it. However, there are also small sugars (mannoses or arabinoses) that surround protein structures yielding additional binding sites according to the P2Rank's criterion (see Supplementary Figure 7). Therefore the total number of binding sites is 29 for the BiteNet's criterion and 38 for the P2Rank's criterion. As a result, BiteNet yields zero false negative predictions in the former case, and 9 in the latter case, hence, the drop in the AP metric from 0.99 to 0.75.

Supporting information:

Binding site definition

P2Rank binding site filtering criterion

This criterion filters small molecules with less than 4 atoms, as well as HOH, DOD, WAT, NAG, MAN, UNK, GLC, ADA, MPD, GOL, SO4, PO4 molecules. The ligand must be within 4Å of a protein, and the distance from the ligand center to protein must be at least 5.5Å. A prediction is considered as true positive, if the minimum distance from the predicted coordinates to the ligand is less or equal than 4Å.

BiteNet binding site filtering criterion

This criterion filters out small molecules with less than 14 number of atoms, as well as detergent and buffer molecules (see Supplementary Table 14). Then binding site interface is defined as the number of protein atoms in the proximity of 4Å from the ligand, and binding sites with the interface less than 20 atoms are filtered out. A prediction is considered as true positive, if the minimum distance between the predicted coordinates and the ligand is less or equal than 4Å.

BiteNet binding site filtering criterion for training

This criterion differs from the previous one only in the true positive definition. Namely, a prediction is considered as true positive, if the minimum distance between the predicted coordinates and the geometric center of the binding site is less or equal than 4Å.

Table 16: The average number of binding sites in the COACH420 and HOLO4K benchmarks according to the P2Rank and BiteNet binding site filtering criteria (the first two rows), as well as the average number of method predictions for each dataset.

Figure 7. Small molecules and BiteNet predictions for 29 proteins of the IPR001461 family from the HOLO4K dataset. A Small molecules corresponding to the relevant binding sites according to both BiteNet and P2Rank filtering criteria (yellow sticks), and only to the P2Rank filtering criterion (magenta sticks). B BiteNet predictions are shown with spheres colored from white to red with respect to the probability score.

2) *Figure 5A shows the average precision for each method and benchmark. It is not clear what this average refers to. There are N known binding sites, a single number of TP and FP for each method, therefore a single precision value per method.*

Answer:

Please see our first comment.

3) *To show the global performance of these methods, Figure 5A should report the Matthews Correlation Coefficient (MCC) of each method instead.*

Answer:

We disagree with the reviewer, as there is no strict definition of the TN metric required to compute MCC (please see our first comment). For example, to define TN for our model one could consider all predictions obtained from each cell ($8 \times 8 \times 8 = 512$ predictions), classify them as T or N w.r.t the probability score threshold of 0.01, and discriminate TP based on the distance threshold to the binding site of 4Å. Thus, TN corresponds to the predictions with the probability score ≤ 0.01 and distance to any true binding site $\geq 4\text{Å}$. The MCC for such a scenario calculated for BiteNet varies from 0.3 to 0.5 on COACH420 and HOLO4K benchmarks, depending on the definition of the binding site. However such MCC cannot be calculated, for example, for fpocket or P2Rank predictions, so we prefer not to include this information into the manuscript, as it could be confusing or misleading to a reader. Note also that none of the methods referred in the manuscript reported the MCC metrics in the corresponding publications, which is probably because of the same reason.

4) In addition to the comparison of MCCs on all proteins, it is important to show the same comparison on each major protein class. Does the ranking of the methods change depending on the protein class?

Answer:

We thank the reviewer for the suggestion to evaluate performance on different protein classes. As MCC is not suitable, we calculated the AP metric on the protein families from the HOLO4K benchmark that have at least 20 structures with at least one binding site. We found that BiteNet outperforms the other methods on 17 out of 27 protein families, for 2 protein families BiteNet is on par with P2Rank showing the perfect performance, and for the rest 8 protein families there is a method with better performance than BiteNet.

Changes in the manuscript:

Added to Section II-C (page 7):

To investigate BiteNet's predictive power in more detail we considered its performance on the most represented protein families comprising the HOLO4K benchmark retrieved from the InterPro database [41]. More precisely, we assigned the InterPro family identifier to each protein in the HOLO4K benchmark and considered protein families counting at least 20 protein structures and containing at least one relevant binding site according to the BiteNet's criterion. Figure 6 shows the AP metric calculated for each protein family for BiteNet, as well as the ratio of structures from this family presented in the training set. BiteNet outperforms the other methods on 17 out of 27 protein families, for 2 protein families BiteNet is on par with P2Rank showing the perfect performance, and for the rest 8 protein families there is a method with better performance than BiteNet (see Supplementary Figure 6). Note also that none of the protein families are over-represented in the training set (the median ratio is 0.15%).

Figure 6. Average precision calculated for protein families with at least 20 protein structures in the HOLO4K benchmark (right side) along with the ratio of structures from each protein family presented in the training set (left side).

Supporting information:

Figure 5: The average precision, precision, and recall metrics for BiteNet with respect to the most represented protein families in the HOLO4K benchmark, along with the ratio of the protein families with respect to the training, validation, and HOLO4K sets.

Figure 6: The average precision metric for different methods with respect to the most represented protein families in the HOLO4K benchmark.

5) Looking at the FPs of BiteNet, what types of proteins have binding sites hard to detect?

6) Looking at the FNs of BiteNet, what types of proteins seem to have binding sites but do not?

Answer:

We thank the reviewer for this note. To demonstrate some of the most frequent types of FP and FN we show an example of Glycosyl transferase protein family in the HOLO4K dataset. The most common false positive predictions correspond to the ligand-free region with low probability score (≤ 0.15). Interestingly, another type of false positive predictions correspond to the region with absent ligand in one structures, but present in the others, so it is not clear whether these predictions should be considered as false positive. At the same time, there are structures with the bound ligand, but with no binding site predictions, corresponding to the most common type of the false negative predictions. Finally, we observed that some false negative predictions correspond to ligands in the proximity of the catalytic binding site predicted with high probability score (≥ 0.75). Such false negative predictions might be an artefact of the non-max-suppression procedure, when only single prediction with the highest probability score is kept within the 8Å.

Changes in the manuscript:

Added to Section II-C (page 8):

Figure 7 demonstrates common types of false positive and false negative predictions on example of Glycosyl transferase protein family (IPR000811). The most common false positive predictions correspond to the ligand-free region with low probability score (≤ 0.15) (see Figure 7A). Interestingly, another type of false positive predictions correspond to the region with absent ligand in one structures, but present in the others (see Figure 7C). Given higher probability scores (≥ 0.20) and capability to bind a ligand to the predicted binding site in some protein structures, it is not clear whether these predictions should be considered as false positive. At the same time, there are structures with the bound ligand, but with no binding site predictions, corresponding to the most common type of the false

negative predictions (see Figure 7B). Finally, we observed that some false negative predictions correspond to ligands in the proximity of the catalytic binding site predicted with high probability score (≥ 0.75) for the PLP molecule (pyridoxal-5'-phosphate) (see Figure 7D). Thus, such false negative predictions might be an artefact of the non-max-suppression procedure, when only single prediction with the highest probability score is kept within the 8Å.

Figure 7. Examples of BiteNet prediction errors on the Glycosyl transferase protein family (IPR000811). A Low scored false positive predictions, for which there are no bound ligands. B False negative prediction, that is absence of predictions in the proximity of the bound ligand. C Similar predictions may correspond to both the true positive and false positive predictions depending on the presence (PDB ID: 3GPB) or absence (PDB ID: 1FU7) of the bound ligand. D Catalytic site with two ligands is predicted either as two (PDB ID: 1LWN) or single (PDB ID: 5GPB) binding sites, resulting in false negative predictions. BiteNet predictions are depicted with spheres colored from white to red with respect to the probability score, and ligands are depicted with magenta, yellow and purple sticks.

7) *BiteNet is compared to other methods on a part of COACH420 and HOLO4K proteins to avoid overlap with training set. I still would like to see the comparison in the full test set (in the supplementary materials).*

Answer:

We extensively evaluated the performance metrics on the entire datasets as well as on its different subsets and observed the same performance trend.

Changes in the manuscript:

In Supporting Information:

Table 1: Performance metrics on the COACH420 subset of 239 proteins, which are not in the BiteNet training set, outputs of all methods are available for them and each structure contains at least one relevant binding site according to the P2Rank binding site filtering criterion.

Table 2: Performance metrics on the HOLO4K subset of 1682 proteins, which are not in the BiteNet training set, outputs of all methods are available for them and each structure contains at least one relevant binding site according to the P2Rank binding site filtering criterion.

Table 3: Performance metrics for the entire COACH420 dataset.

Table 4: Performance metrics for the entire HOLO4K dataset.

Table 5: Performance metrics on the COACH420 subset of 352 proteins, which are not in the BiteNet training set and each structure contains at least one relevant binding site according to the P2Rank binding site filtering criterion.

Table 6: Performance metrics on the HOLO4K subset of 4169 proteins, which are not in the BiteNet training set and each structure contains at least one relevant binding site according to the P2Rank binding site filtering criterion.

Table 7: Performance metrics on the COACH420 subset of 344 proteins, which are not in both BiteNet training and validation set and each structure contains at least one relevant binding site according to the P2Rank binding site filtering criterion.

Table 8: Performance metrics on the HOLO4K subset of 4121 proteins, which are not in BiteNet training and validation sets and each structure contains at least one relevant binding site according to the P2Rank binding site filtering criterion.

Table 9: Performance metrics on the COACH420 subset of 238 proteins, which are not in the BiteNet training set and each structure contains at least one relevant binding site according to the P2Rank binding site filtering criterion.

Table 10: Performance metrics on the HOLO4K subset of 2848 proteins, which are not in the BiteNet training set and each structure contains at least one relevant binding site according to the BiteNet binding site filtering criterion.

8) The authors said “We trained several models and found the following hyperparameters to be optimal”. I found this worrying: where they optimal because they provide the best precision on the test sets? If so, BiteNet performance on those sets would be overestimated (an unfair comparison).

Answer:

We determined optimal hyperparameters with respect to the validation set, and not to the test benchmarks. The intersection between the test benchmarks and the training set is zero. The intersection between the test benchmarks and the validation set is 8 and 48 structures for the COACH420 and HOLO4K benchmarks, respectively. We also calculated performance on the

COACH420 and HOLO4K benchmarks, excluding these 8 and 48 structures, so there is no intersection at all - the observed difference is only in the third decimal place.

Changes in the manuscript:

Tables 1-10 in Supporting information

REVIEWERS' COMMENTS:

Reviewer #1 (Remarks to the Author):

The authors have addressed all my concerns. Therefore, I recommend it for the publication.

Reviewer #2 (Remarks to the Author):

No additional comments

Reviewer #3 (Remarks to the Author):

The authors have clarified some of the parts that were obscure in the original version.

They however refused to provide the requested comparisons by making a number of unsupported claims: the requested metrics can be easily overtrained (but do not say how), precision and recall "tends" to be higher with smaller and larger number of predicted binding sites (doubtful, especially given small differences of this variable across proteins), AP is however suitable (again no proof provided).

Some of these justifications are obviously incorrect: there is not an infinite number of points around the protein structure surface that can be classified as not a binding point. Only convex cavities can be considered small-molecule binding sites and, given that interatomic distances are not smaller than 1Å, it does not make sense to consider smaller numerical precisions than that. Thus, this number is not only finite, but relatively small.

These responses show a lack of understanding of the subtleties of this problem. The definition of a TN is actually as straightforward as that of a TP. Once all the convex cavities of a protein have been determined, those ligand-bound are the binding sites and those without a ligand are non-binding sites. Of course, another structure of the same, or structurally similar, protein might have that "non-binding" pocket ligand-bound. But this is equally true for the binding site, which might not be ligand bound in another structure. A proper benchmark for this problem must consider as binding site any cavity that is ligand-bound in at least one structure of that protein. Likewise, a non-binding site would be those that are not ligand-bound in *none of the structures of that target*.

In conclusion, these serious shortcomings do not allow me to support the publication of this study.

Reviewer 2 comments on Reviewer 3's concerns (Remarks to Editor):

I had a look at the reviewer's comments you attached.
Please find my comments below, on a point-by-point basis.

< "They however refused to provide the requested comparisons by making a number of unsupported claims: the requested metrics can be easily overtrained (but do not say how), precision and recall "tends" to be higher with smaller and larger number of predicted binding sites (doubtful, especially given small differences of this variable across proteins), AP is however suitable (again no proof provided)."

<< I agree with the reviewer that the authors failed to explain their decision to not include the requested comparison. However, what the authors say regarding precision and recall is true. Precision decreases when the number of predicted binding sites increases (as there is an increasing risk to have FP at the denominator), while recall increases (decreased number of FN at

the denominator). These are only generic trends, that should be supported on a case-by-case basis and ultimately also depend on how good the underlying models are. The authors could have provided a better explanation in their response letter, but the metrics they chose are not wrong by default.

<

< "Some of these justifications are obviously incorrect: there is not an infinite number of points around the protein structure surface that can be classified as not a binding point. Only convex cavities can be considered small-molecule binding sites and, given that interatomic distances are not smaller than 1Å, it does not make sense to consider smaller numerical precisions than that. Thus, this number is not only finite, but relatively small."

<< I actually agree with the authors, there is indeed an infinite number of points around a surface that are not a binding point, as we are looking at a "real" system. As such, the system could be theoretically described by an infinite amount of points in the three-dimensional space. In my opinion, this looks more like a philosophical discussion rather than something that really affects the scientific soundness of the paper.

< These responses show a lack of understanding of the subtleties of this problem. The definition of a TN is actually as straightforward as that of a TP. Once all the convex cavities of a protein have been determined, those ligand-bound are the binding sites and those without a ligand are non-binding sites. Of course, another structure of the same, or structurally similar, protein might have that "non-binding" pocket ligand-bound. But this is equally true for the binding site, which might not be ligand bound in another structure. A proper benchmark for this problem must consider as binding site any cavity that is ligand-bound in at least one structure of that protein. Likewise, a non-binding site would be those that are not ligand-bound in *none of the structures of that target*.

<< I understand the reviewer's comment. I would also have tried to find a way to capture true negatives in a more straightforward way. Nonetheless, even the solution proposed by the reviewer can be subjected to artefacts, due to only a partial knowledge of which binding sites are truly druggable. In my opinion, as far as all the methods are compared in a similar way, with the metrics unambiguously defined, the conclusions of the study are verifiable and limited to what was captured by those metrics. Thus, once again, despite this part could have been performed in a different way, I do not think that the approach itself lacks the basic scientific soundness needed for publication.

Thus, all in all, I don't feel like the aspects highlighted by the reviewer (all relative to the same point) constitute a sufficient reason for paper rejection.

Thank you for taking your time to consider my opinion; I hope my feedback was useful for your decision.